# DISCOVERING DISTINCTIVE "SEMANTICS" IN SUPER-RESOLUTION NETWORKS

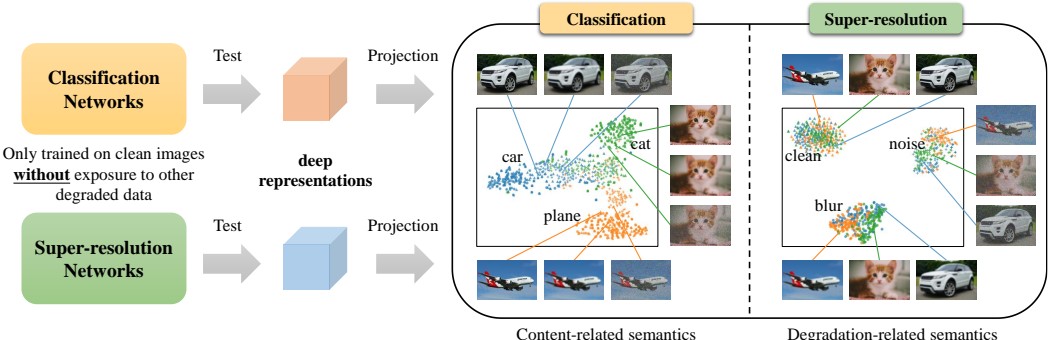

Figure 1: Distributions of the deep representations of classification and super-resolution networks. For classification networks, the semantics of the deep feature representations are artificially predefined according to the training data (category labels). However, for SR networks, the learned deep representations have a different kind of "semantics" from classification. During training, the SR networks are only provided with downsampled clean LR images. There is not any supervision signal related to image degradation information. Surprisingly, we find that SR networks' deep representations are spontaneously discriminative to different degradations. Notably, NOT an arbitrary SR network has such a property. In Sec. 4.3, we reveal two factors that facilitate SR networks to extract such degradation-related representations, i.e., adversarial learning and global residual.

## ABSTRACT

Image super-resolution (SR) is a representative low-level vision problem. Although deep SR networks have achieved extraordinary success, we are still unaware of their working mechanisms. Specifically, whether SR networks can learn semantic information, or just perform complex mapping functions? What hinders SR networks from generalizing to real-world data? These questions not only raise our curiosity, but also influence SR network development. In this paper, we make the primary attempt to answer the above fundamental questions. After comprehensively analyzing the feature representations (via dimensionality reduction and visualization), we successfully discover the distinctive "semantics" in SR networks, i.e., deep degradation representations (DDR), which relate to image degradation instead of image content. We show that a well-trained deep SR network is naturally a good descriptor of degradation information. Our experiments also reveal two key factors (adversarial learning and global residual) that influence the extraction of such semantics. We further apply DDR in several interesting applications (such as distortion identification, blind SR and generalization evaluation) and achieve promising results, demonstrating the correctness and effectiveness of our findings.

## 1 INTRODUCTION

The emergence of deep convolutional neural network (CNN) has given birth to a large number of new solutions to low-level vision tasks (Dong et al., 2014; Zhang et al., 2017). Among these signs of progress, image super-resolution (SR) has enjoyed a great performance leap. Compared with traditional methods (e.g., interpolation (Keys, 1981) and sparse coding (Yang et al., 2008)), SR networks can achieve better performance with improved efficiency.

However, even if we have benefited a lot from the powerful CNNs, we have little knowledge about what happens in SR networks and what distinguishes them from traditional approaches on earth. Does the performance gain merely come from more complex mapping functions? Or is there anything different inside SR networks, like classification networks with discriminative capability? On the other hand, as a classic regression task, SR is expected to perform a continuous mapping from low-resolution (LR) to high-resolution (HR) images. It is generally a local operation without the consideration of the global context. But with the introduction of GAN-based models Ledig et al. (2017); Wang et al. (2018), more delicate SR textures can be generated. It seems that the network has learned some kind of semantic, which is beyond our common perception for regression tasks.

Then, we may raise the question: are there any "semantics" in SR networks? If yes, do these semantics have different definitions from those in classification networks? Existing literature cannot answer these questions, as there is little research on interpreting low-level vision deep models. Nevertheless, discovering the semantics in SR networks is of great importance. It can not only help us further understand the underlying working mechanisms, but also guide us to design better networks and evaluation algorithms.

In this study, we give affirmative answers to the above questions by unfolding the semantics hidden in super-resolution networks. Specifically, different from the artificially predefined semantics associated with object classes in high-level vision, semantics in SR networks are distinct in terms of ***image degradation*** instead of image content. Accordingly, we name such semantics deep degradation representations (DDR). More interestingly, such degradation-related semantics are spontaneously existing without any predefined labels. We reveal that **a well-trained deep SR network is naturally a good descriptor of degradation information.**

Notably, the semantics in this paper have different implications from those in high-level vision. Previously, researchers have disclosed the hierarchical nature of classification networks (Zeiler & Fergus, 2014; Gu et al., 2018). As the layer deepens, the learned features respond more to abstract high-level patterns (*e.g.*, faces and legs), showing a stronger discriminability to object categories (see Fig. 4). However, similar research in low-level vision is absent, since there are no predefined semantic labels. In this paper, we reveal the differences in deep "semantics" between classification and SR networks, as illustrated in Fig. 1.

Our observation stems from a representative blind SR method – CinCGAN Yuan et al. (2018), and we further extend it to more common SR networks – SRResNet and SRGAN Ledig et al. (2017). We have also revealed more interesting phenomena to help interpret the semantics, including the analogy to classification networks and the influential factors for extracting DDR. Moreover, we improve the results of several tasks by exploiting DDR. We believe our findings could lay the groundwork for the interpretability of SR networks, and inspire more exploration of the mechanism of low-level vision deep models.

**Contributions.** 1) We have successfully discovered the "semantics" in SR networks, denoted as deep degradation representations (DDR). Through in-depth analysis, we also find that global residual learning and adversarial learning can facilitate the SR network to extract such degradation-related representations. 2) We reveal the differences in deep representations between classification and SR networks, for the first time. This further expands our knowledge of the deep representations of high- and low-level vision models. 3) We exploit our findings to several fundamental tasks and achieve very appealing results, including distortion identification, blind SR and generalization evaluation.

## 2  RELATED WORK

**Super-resolution.** Super-resolution (SR) is a fundamental task in low-level vision, which aims to reconstruct the high-resolution (HR) image from the corresponding low-resolution (LR) counterpart. SRCNN (Dong et al., 2014) is the first proposed CNN-based method for SR. Since then, a large number of deep-learning-based methods have been developed (Dong et al., 2016; Lim et al., 2017; Zhang et al., 2018b; Ledig et al., 2017; Zhang et al., 2019). Generally, current CNN-based SR methods can be categorized into two groups. One is MSE-based method, which targets at minimizing the distortion (e.g., Mean Square Error) between the ground-truth HR image and super-resolved image to yield high PSNR values, such as SRCNN (Dong et al., 2014), VDSR (Kim et al., 2016), EDSR (Lim et al., 2017), RCAN (Zhang et al., 2018b), SAN (Dai et al., 2019), etc. The other is GAN-based method, which incorporates generative adversarial network (GAN) and perceptual loss (Johnson et al., 2016) to obtain perceptually pleasing results, such as SRGAN (Ledig et al., 2017),

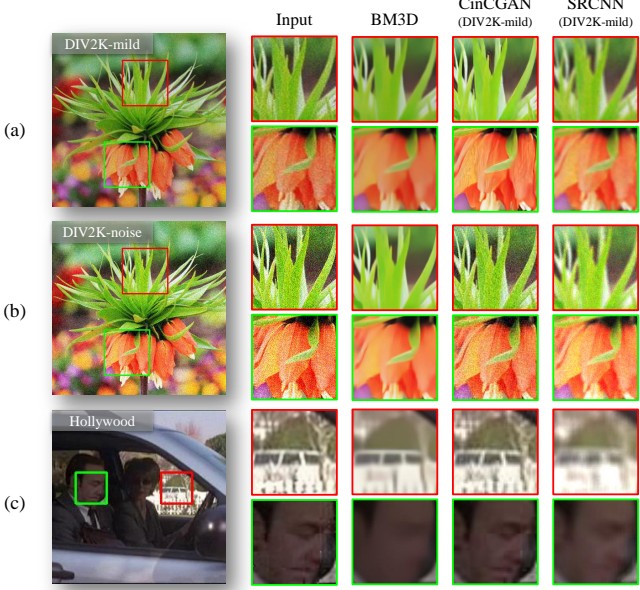

Figure 2: Different degraded input images and their corresponding outputs produced by CinCGAN (Yuan et al., 2018), BM3D (Dabov et al., 2007), and SRCNN (Dong et al., 2014). CinCGAN (Yuan et al., 2018) is trained on DIV2K-mild dataset in an unpaired manner. If the input image conforms to the training data distribution, CinCGAN will generate better restoration results than BM3D (a). Otherwise, it tends to ignore the unseen degradation types (b)&(c). On the other hand, the traditional method BM3D (Dabov et al., 2007) has stable performance and similar denoising effects on all input images, regardless of the input degradation types. Zoom in for the best view.

ESRGAN (Wang et al., 2018), RankSRGAN (Zhang et al., 2019), SROBB (Rad et al., 2019). Recently, blind SR has attracted more and more attention (Gu et al., 2019; Bell-Kligler et al., 2019; Luo et al., 2020; Wang et al., 2021),which aims to solve SR with unknown real-world degradation. A comprehensive survey for blind SR is newly proposed (Liu et al., 2021), which summarizes existing methods. We regard SR as a representative research object and study its deep semantic representations. It can also draw inspirations on other low-level vision tasks.

**Network interpretability.** At present, most existing works on neural network interpretability focus on high-level vision tasks, especially for image classification. Zhang *et al*. (Zhang et al., 2020) systematically reviewed existing literature on network interpretability and proposed a novel taxonomy to categorize them. Here we only discuss several classic works. By adopting deconvolutional networks (Zeiler et al., 2010), Zeiler *et al*. (Zeiler & Fergus, 2014) projected the downsampled low-resolution feature activations back to the input pixel space, and then performed a sensitivity analysis to reveal which parts of the image are important for classification. Simonyan *et al*. (Simonyan et al., 2013) generated a saliency map from the gradients through a single backpropagation pass. Based on class activation maps (CAM) (Zhou et al., 2016), Selvaraju *et al*. (Selvaraju et al., 2017) proposed Grad-CAM (Gradient-weighted CAM) to produce a coarse-grained attribution map of the important regions in the image, which was broadly applicable to any CNN-based architecture. For more information about the network interpretability literature, please refer to the survey paper (Zhang et al., 2020). However, for low-level vision tasks, similar researches are rare. Recently, the local attribution map (LAM) (Gu & Dong, 2021) has been proposed to interpret super-resolution networks, which can be used to localize the input features that influenced the network outputs. Besides, Wang *et al*. (Wang et al., 2020b) presented a pioneer work that bridges the representation relationship between high- and low-level vision. They learned the mapping between deep representations of low-and high-quality images, and leveraged it as a deep degradation prior (DDP) for low-quality image classification. Inspired by these previous works, we interpret SR networks from another new perspective. We dive into their deep feature representations, and discover the "semantics" of SR networks. More background knowledge is described in the supplementary file.

# 3 MOTIVATION

To begin with, we present an interesting phenomenon, which drives us to start exploring the deep representations of SR networks. It is well known that SR networks are superior to traditional methods in specific scenarios, but are inferior in generalization ability. In blind SR, the degradation types of the input test images are unknown. For traditional methods, they treat different images equally without distinction of degradation types, thus their performance is generally stable and predictable. How about the SR networks, especially those designed for blind SR?

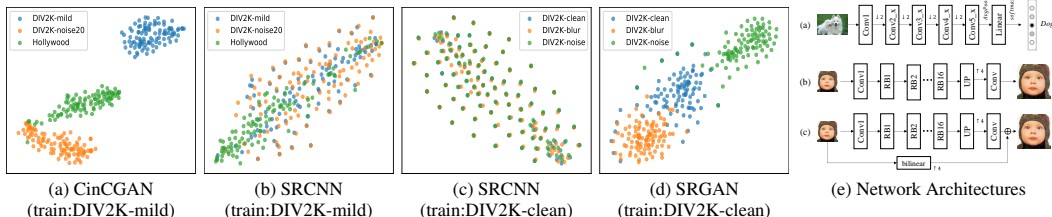

(a) CinCGAN    (b) SRCNN    (c) SRCNN    (d) SRGAN    (e) Network Architectures
(train:DIV2K-mild) (train:DIV2K-mild) (train:DIV2K-clean) (train:DIV2K-clean)

Figure 3: (a)-(d): The projected deep feature representations. The deep features of CinCGAN and SRGAN are separated by degradation types, even if the image contents are aligned. (e)-a: ResNet18 (He et al., 2016) for classification. "Conv2_x" represents the 2nd group of residual blocks. (e)-b: SRResNet-woGR (without global residual). (e)-c: SRResNet (with global residual). "RB1" represents the 1st residual block.

CinCGAN (Yuan et al., 2018) is a representative solution for real-world SR without paired training data. It maps a degraded LR to its clean version using data distribution learning before conducting SR operation. However, we find that it still has a limited application scope even if CinCGAN is developed for blind settings. If the degradation of the input image is not included in the training data, CinCGAN will fail to transfer the degraded input to a clean one. More interestingly, instead of producing extra artifacts in the image, it seems that CinCGAN does not process the input image and retains all the original defects. Readers can refer to Fig. 2 for an illustration, where CinCGAN performs well on the testing image of the DIV2K-mild dataset (same distribution as its training data), but produces unsatisfactory results for other different degradation types. In other words, *the network seems to figure out the specific degradation types within its training data distribution, and distribution mismatch may make the network "turn off" its ability.* This makes the performance of CinCGAN unstable and unpredictable. For comparison, we process the above three types of degraded images by a traditional denoising method BM3D (Dabov et al., 2007) [1]. The visual results show that BM3D has an obvious and stable denoising performance for all different degradation types. Although the results of BM3D may be mediocre (the image textures are largely over-smoothed), it does take effect on every input image. This observation reveals a significant discrepancy between traditional methods and SR networks.

The above interesting phenomenon indicates that the deep network has learned more than a regression function, since it demonstrates the ability to distinguish among different degradation types. Inspired by this observation, we try to find any semantics hidden in SR networks.

## 4 DIVING INTO THE DEEP DEGRADATION REPRESENTATIONS

### 4.1 DISCRIMINABILITY OF DEEP REPRESENTATIONS IN DEEP SR NETWORKS

**Feature projection and visualization.** Since the final outputs are always derived from features in CNN layers, we start the exploration with feature maps, especially the deep ones potentially with more global and abstract information. To interpret the deep features of CNN, one common and rational way is to convert the high-dimensional CNN feature maps into lower-dimensional datapoints that can be visualized in a scatterplot. Afterwards, one can intuitively understand the data structures and manifolds. Specifically, we adopt t-Distributed Stochastic Neighbor Embedding (t-SNE) (Van der Maaten & Hinton, 2008) for dimensionality reduction. This algorithm is commonly used in manifold learning, and it has been successfully applied in previous works (Donahue et al., 2014; Mnih et al., 2015; Wen et al., 2016; Zahavy et al., 2016; Veličković et al., 2017; Wang et al., 2020b; Huang et al., 2020) for feature projection and visualization. In our experiments, we first reduce the dimensionality of feature maps to a reasonable amount (50 in this paper) using PCA (Hotelling, 1933), then apply t-SNE to project the 50-dimensional representation to two-dimensional space, after which the results are visualized in a scatterplot. Furthermore, we also introduce CHI (Caliński & Harabasz, 1974) score to quantitatively evaluate the distributions of visualized datapoints. The CHI score is higher when clusters are well separated, which indicates stronger semantic discriminability.

**What do the deep features of SR networks represent?** As discussed in Sec.3, since CinCGAN performs differently on various degradations, we compare the features generated from three testing datasets: 1) DIV2K-mild: training and testing data used in CinCGAN, which are synthesized

---

[1]Note that BM3D is a denoising method while CinCGAN is able to upsample the resolution of the input image. Thus, after applying BM3D, we apply bicubic interpolation to unify the resolution of the output image. This is reasonable as we only evaluate their denoising effects.

from DIV2K (Agustsson & Timofte, 2017) dataset, containing noise, blur, pixel shifting and other degradations. 2) DIV2K-noise20: add Gaussian noise ($\sigma = 20$) to DIV2K set. 3) Hollywood100: 100 images selected from Hollywood dataset (Laptev et al., 2008), containing real-world old film degradations. Each test dataset includes 100 images.

As shown in Fig. 3(a), there is a strong feature discriminability for various degradations. Images with aligned contents but different degradation types are still separated into different clusters. [2] This phenomenon conforms to our observation that CinCGAN *does* treat various input degradations in different ways. It naturally reveals the "semantics" of deep representations in CinCGAN, which are closely related to the degradation types rather than the image content. For comparison, we may wonder whether traditional methods have similar behaviors (or "semantics"). However, our feature analysis method can only work for deep models, which contain hierarchical feature maps. It is acknowledged that the simplest network – SRCNN can be analogous to a sparse-coding-based method, thus we can use SRCNN to shed light on the behaviors of traditional methods. We train an SRCNN[3] with the same data as CinCGAN, and visualize the feature representations of the last layer in Fig. 3(b). It is obvious that different degradations cannot be clearly separated. This phenomenon is completely different from CinCGAN. We can conjecture that the degradation-related semantics only exist in deep models, not traditional methods or shallow networks. More analysis on shallow networks can be found in the supplementary file.

**From CinCGAN to Generic SRGAN.** Notably, the training of CinCGAN involves degraded images (DIV2K-mild). It actually performs simultaneous restoration and SR. We also wonder how this kind of degradation-related semantics manifests in classical SR networks (without exposure to other degradation types except for downsampling). Therefore, we adopt a generic GAN-based SR network SRGAN (Ledig et al., 2017; Wang et al., 2018) to conduct the visualization experiment. SRGAN is trained with DIV2K dataset (Agustsson & Timofte, 2017) with **only** bicubic-downsampled LR images. According to the common degradation modelling in low-level vision, we use three datasets with different degradation types for testing: 1) DIV2K-clean: the original DIV2K validation set containing only bicubic downsampling degradation, which conforms to the training data distribution. 2) DIV2K-blur: introduce blurring degradation with Gaussian blur kernel on the DIV2K-clean set. The kernel width is randomly sampled from $[2, 4]$ for each image and the kernel size is fixed to $15 \times 15$. 3) DIV2K-noise: add Gaussian noises to the DIV2K-clean set. The noise level is randomly sampled from $[5, 30]$ for each image. These three testing datasets are aligned in image content but different in degradation types.

As shown in Fig.3(d), a clustering trend similar to CinCGAN is clearly demonstrated. This provides stronger evidence for the existence of degradation-related semantics. Even if the three testing sets share the same content, they are still separated into distinct clusters according to the degradation types. In the supplementary file, similar phenomena are observed with other network structures. Note again, shallow SRCNN does not have such feature discriminability (see Fig.3(c)).

There, we successfully find the semantics hidden in deep SR networks. They are perceivable to humans when visualized in low-dimensional space. Specifically, ***semantics in deep SR networks are in terms of degradation types regardless of the image contents***. Simply but vividly, we name this kind of semantics as deep degradation representations (DDR).

**Is DDR a natural and trivial observation?** No, there are three reasons. First, DDR has never been discussed before. The function of deep SR networks is beyond simple regression. The learned deep features can spontaneously characterize the image degradations, indicating that *a well-trained deep SR network is naturally a good descriptor of degradation information*. Note again that the deep SR networks have not observed any blurry or noisy data during training, but still have the discriminative ability on different degradations. Second, DDR in SR is *not* simply caused by different input patterns. We find that different networks will learn different semantic representations. For example, in Sec. 4.2, we reveal the differences in the learned representations between classification and SR Networks. In Sec. 4.3, we show that not all SR network structures can easily obtain DDR. DDR does not exist in specific cases and shallow networks. Third, DDR has important applications and inspirations. It can not only expand our understanding of the underlying mechanisms of low-level

---

[2] Note that the class labels in the scatterplots are only used to assign a color/symbol to the datapoints for better visualization.

[3] We use the same architecture as the original paper Dong et al. (2014) and add global residual for better visualization.

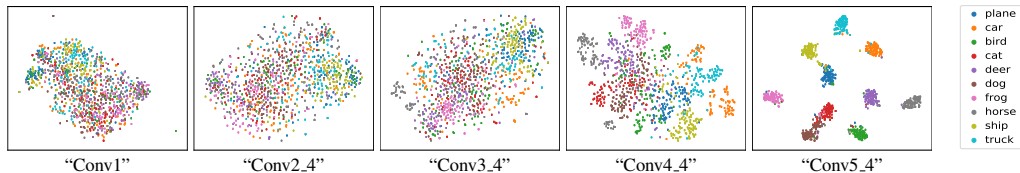

Figure 4: Projected feature representations extracted from different layers of ResNet18 using t-SNE. With the network deepens, the representations become more discriminative to object categories, which clearly shows the semantics of the representations in classification.

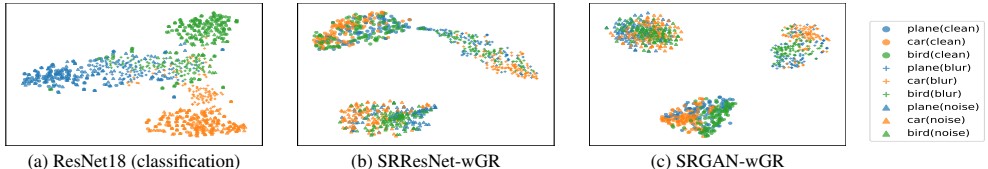

Figure 5: Feature representation differences between classification and SR networks. The same object category is represented by the same color, and the same image degradation type is depicted by the same marker shape. For the classification network, feature representations are clustered by the same color, while representations of SR networks are clustered by the same marker shape, suggesting that there is a significant difference in feature representations between classification and SR networks.

vision models, but also help promote the development of other tasks. In Sec. 5, we apply DDR to several fundamental tasks and achieve appealing results, implying the great potential of DDR.

## 4.2 DIFFERENCES IN SEMANTICS BETWEEN CLASSIFICATION AND SR NETWORKS

In the high-level vision, classification is one of the most representative tasks, where artificially pre-defined semantic labels on object classes are given as supervision. We choose ResNet18 (He et al., 2016) as the classification backbone and conduct experiments on CIFAR10 dataset (Krizhevsky et al., 2009). We extract the forward features of each input testing image[4] at different network layers, as described in Fig. 3(e)-a.

Fig. 4 shows that as the network deepens, the extracted feature representations produce obvious discriminative clusters, i.e., the learned features are increasingly becoming semantically discriminative. Such discriminative *semantics in classification networks are coherent with the artificially predefined labels*. This is an intuitive and natural observation, on which lots of representation and discriminative learning methods are based (Wen et al., 2016; Oord et al., 2018; Lee et al., 2019; Wang et al., 2020b).

Further, we add blur and noise degradation to the CIFAR10 test images, and then investigate the feature representations of classification and SR networks. Note that no degradation is added to the training data. As shown in Fig. 5, after adding degradations to the test data, the deep representations obtained by the classification network (ResNet18) are still clustered by object categories, indicating that the features focus more on high-level object class information. On the contrary, the deep representations obtained by SR networks (SRResNet and SRGAN) are clustered with regard to degradation types. The features of the same object category are not clustered together, while those of the same degradation type are clustered together, showing different "semantic" discriminability. This phenomenon intuitively illustrates the differences in the deep semantic representations between SR and classification networks, i.e., degradation-related semantics and content-related semantics. More interestingly, the "semantics" in SR networks exists naturally, because the SR networks only see clean data without any input or labelled degradation information.

## 4.3 HOW DO GLOBAL RESIDUAL AND ADVERSARIAL LEARNING AFFECT THE DEEP REPRESENTATIONS?

Previously, we have elaborated on the deep degradation representations in CinCGAN, SRGAN and SRResNet. Nevertheless, we further discover that no arbitrary SR network structure has such a property. To be specific, we find two crucial factors that can influence the learned representations: i) image global residual (GR), and ii) generative adversarial learning (GAN).

---

[4]For efficiency, we selected 100 testing images of each category (1000 images in total).

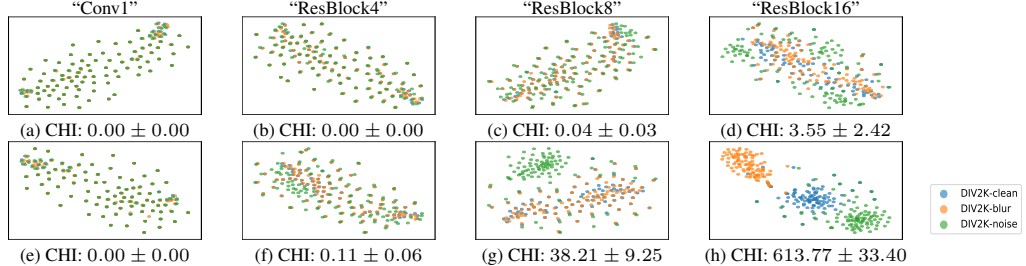

Figure 6: Projected feature representations extracted from different layers of SRResNet-woGR (1st row) and SRResNet (2nd row) using t-SNE. With image global residual (GR), the representations of MSE-based SR networks show discriminability to degradation types.

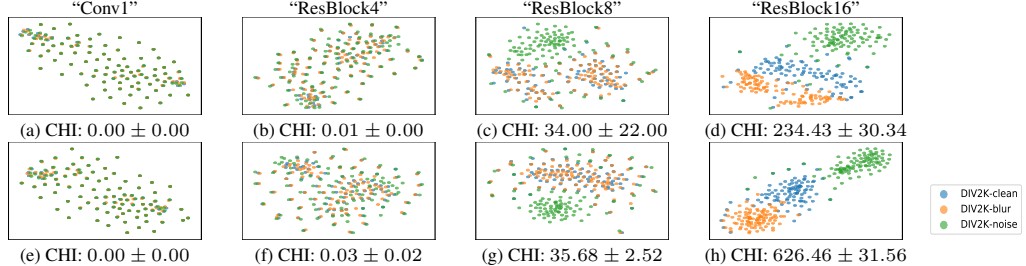

Figure 7: Projected feature representations extracted from different layers of SRGAN-woGR (1st row) and SRGAN (2nd row) using t-SNE. Even without GR, GAN-based SR networks can still obtain DDR.

**Global Residual.** We train two SRResNet networks – SRResNet (with global residual) and SRResNet-woGR (without global residual), as shown in Fig. 3. The two architectures are both common in practice (Kim et al., 2016; Shi et al., 2016). DIV2K (Agustsson & Timofte, 2017) dataset is used for training, where the LR images are bicubic-downsampled and clean. Readers can refer to the supplementary file for more details. After testing, the feature visualization analysis is shown in Fig. 6.

The results show that for MSE-based SR method, GR is essential for producing discriminative representations on degradation types. The features in "ResBlock16" of SRResNet have shown distinct discriminability, where the clean, blur, and noise data are clustered separately. On the contrary, SRResNet-woGR shows no discriminability even in deep layers. This phenomenon reveals that GR significantly impacts the learned feature representations. It is inferred that learning the global residual could remove most of the content information and make the network concentrate more on the contained degradation. This claim is also corroborated by visualizing the feature maps in the supplementary file.

**Adversarial Learning.** MSE-based and GAN-based methods are currently two prevailing trends in CNN-based SR methods. Previous studies only reveal that the output images of MSE-based and GAN-based methods are different, but the differences between their feature representations are rarely discussed. Since their learning mechanisms are quite different, will there be a discrepancy in their deep feature representations? We directly adopt SRResNet and SRResNet-woGR as generators. Consequently, we build two corresponding GAN-based models, namely SRGAN and SRGAN-woGR. After training, we perform the same test and analysis process mentioned earlier.

The results show that the deep features are bound to be discriminative to degradation types for the GAN-based method, whether there is GR or not. As shown in Fig. 7(d)(h), the deep representations in "ResBlock16" of SRGAN-woGR have already been clustered according to different degradation types. This suggests that the learned deep representations of MSE-based method and GAN-based method are dissimilar. Adversarial learning can help the network learn more informative features for distinguishing image degradation rather than image content.

### 4.4 How Does DDR Evolve Through the Training Process?

We also reveal the relationship between the model performance and DDR discriminability. We select SRResNet models with different training iterations for testing. We report the model performance

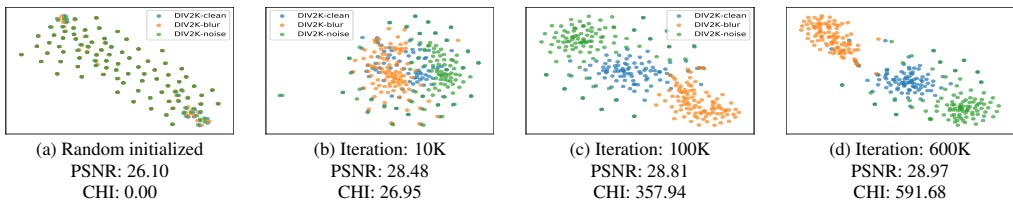

(a) Random initialized
PSNR: 26.10
CHI: 0.00

(b) Iteration: 10K
PSNR: 28.48
CHI: 26.95

(c) Iteration: 100K
PSNR: 28.81
CHI: 357.94

(d) Iteration: 600K
PSNR: 28.97
CHI: 591.68

Figure 8: As the training process goes, the performance and discriminability improve simultaneously.

Table 1: Distortion identification accuracy. (a) BRISQUE (b) MLLNet-2 + PA.

|  | GB | WN | JPEG | JP2K | FF | ALL |
|---|---|---|---|---|---|---|
| (a) | 0.97 | 1.00 | 0.89 | 0.83 | 0.84 | 0.89 |
| (b) | – | – | – | – | – | 0.91 |
| DDR | **0.97** | **1.00** | **1.00** | **0.98** | **0.88** | **0.96** |

Table 2: The PSNR↑/NIQE↓ results on datasets with different degradations.

|  | Blur2 | Noise20 | B2+N20 |
|---|---|---|---|
| DASR (b+n) | 23.28/6.74 | 22.23/7.05 | 21.32/7.53 |
| IKC (b) | 23.74/6.56 | 16.60/7.22 | 16.19/6.87 |
| MANet (b+n) | 15.97/6.37 | 16.32/**6.61** | 16.83/**7.29** |
| DAN (b+n) | **23.94**/6.44 | 18.46/8.20 | 17.76/8.04 |
| RRDB (clean) | 21.40/8.01 | 17.80/8.29 | 17.23/8.73 |
| RRDB (b+n) | 23.79/**6.36** | **22.54**/6.66 | **21.36**/7.36 |
| RRDB-DDR (b+n) | **24.01**/**6.34** | 22.52/**6.60** | **21.41**/**7.27** |

on DIV2K-clean validation dataset and calculate the CHI scores to evaluate its discriminability with clean, blur and noise data. As shown in Fig. 8, as the training process goes, the performance of the model is improved, while the feature discriminability for degradation is also enhanced. From random initialization to 700k iterations, the CHI score increases significantly from 0.00 to 591.68, while the PSNR value improves by 2.87dB (Due to GR, the initial PSNR value is relatively high). The training data only include clean LR images, but the trained model has the ability to discriminate unseen degradation types. This clearly implies that a well-trained deep SR network is naturally a good descriptor of degradation information.

### 4.5 Further Discussion on the Causes of DDR Phenomenon

In the previous sections, we reveal several important factors that promote the manifestation of DDR phenomenon, including global residual, adversarial learning (Sec. 4.3) and training iterations (Sec. 4.4). Based on the above findings and more visualization results, we can analyze the causes of DDR more deeply. We visualize the feature maps of SRResNet-wGR, SRResNet-woGR, SRGAN-wGR and SRGAN-woGR on test images with different degradations in the Appendix.

The DDR phenomenon is mainly introduced by *overfitting* the degradation in the training data. Specifically, since the training data (DIV2K-clean) do not contain extra degradations, the trained SR network lacks the ability to deal with the unseen degradations. When feeding images with degradations (e.g., noise and blur), it will produce features with unprocessed noises or blurring. These patterned features naturally show a strong discriminability between different degradations. As for GR, models with GR produce features that contain less components of original content information. GR can help remove the redundant image content information and make the network concentrate more on degradation-related information. GAN training also enhances the high-frequency degradation information. Besides, prolonging the training iterations and deepening the network depth will make the network further overfit to the training data.

### 4.6 Why SR Networks Can Hardly Generalize to Unseen Degradations?

Classical SR models (Dong et al., 2014; Lim et al., 2017) assume that the input LR images are generated by fixed downsampling kernel (*e.g.*, bicubic). However, it is difficult to apply such simple SR models to real scenarios with unknown degradations. We claim that SR and restoration networks learn to overfit the distribution of degradations, rather than the distribution of natural clean images.

To verify our statements, we compare the representations between SRGAN-wGR models trained on clean data and clean+noise data, respectively. As presented in Fig. 9, if the model is trained only on clean LR data, the deep representations show strong discriminability to clean and noise data. In contrast, if the model sees noise data during training, such discriminability diminishes. The model will become more robust to more degradation types, as the distributions of the deep representations become unanimous. In summary, to improve the model generalization for various degradations, we need to diminish the feature discriminability to degradations. Adding more degraded data into training is a plausible way to enhance the generalization.

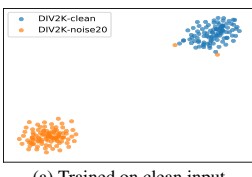 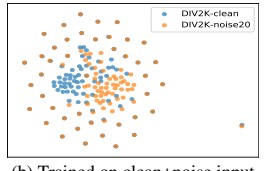

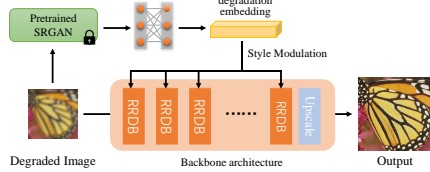

(a) Trained on clean input    (b) Trained on clean+noise input

Figure 9: By training with more degraded data, the deep representations become unanimous.

Figure 10: RRDBNet with DDR guidance. The degradation embedding is injected into the backbone network.

## 5 APPLICATIONS AND INSPIRATIONS

**Image Distortion Identification Using DDR Features.** Image distortion identification (Liang et al., 2020) is an important subsidiary pretreatment for many image processing systems, especially for image quality assessment (IQA). It aims to recognize the distortion type from the distorted images, so as to facilitate the downstream tasks (Mittal et al., 2012a; Gu et al., 2019; Liang et al., 2020). Previous methods usually resort to design handcrafted features that can distinguish different degradation types (Mittal et al., 2012a;b) or train a classification model via supervised learning (Kang et al., 2014; Bosse et al., 2017; Liang et al., 2020). Since DDR is related to image degradation, it can naturally be used as an excellent prior feature for image distortion identification. To obtain DDR, we do not need any degradation information but only a well-trained SR model (train on clean data). Following BRISQUE (Mittal et al., 2012a), we adopt the deep representations of SRGAN as input features (using PCA to reduce the original features to a 120-dimensional vector), and then use linear SVM to classify the degradation types of LIVE dataset (Sheikh et al., 2006). As shown in Tab. 1, compared with BRISQUE and MLLNet (Liang et al., 2020), DDR features achieve excellent results on recognizing different distortion types. More inspiringly, DDR is not obtained by any distortion-related supervision.

**Blind SR with DDR Guidance.** To super-resolve real images with unknown degradations, many blind SR methods resort to estimating and utilising the degradation information. For instance, IKC (Gu et al., 2019) iteratively corrects the estimated blur kernel, and DASR (Wang et al., 2021) implicitly learns the degradation representations by contrastive learning. Based on the findings of DDR, we adopt a trained SRGAN model to extract degradation embedding to promote blind SR models. RRDBNet (Wang et al., 2018) is adopted as the backbone. The DDR embedding is injected into each RRDB module by the StyleMod Karras et al. (2020) (see Fig. 10). The training data are described in Tab. 2, *e.g.*, "b+n" means that the training data include blur and noise images. DDR guidance can help improve the model performance. Fig. 11 reveals that DDR guidance can make the deep features become more homogeneous (CHI scores drop from 14.04 to 4.95).

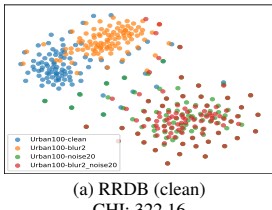 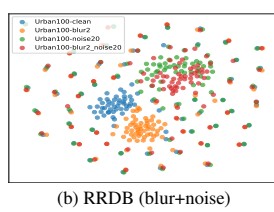 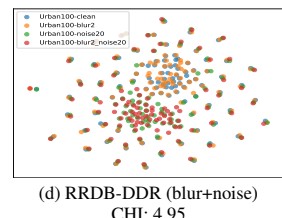

(a) RRDB (clean)        (b) RRDB (blur+noise)        (d) RRDB-DDR (blur+noise)
CHI: 322.16               CHI: 14.04               CHI: 4.95

Figure 11: DDR clustering of different models. A lower CHI score connotes better generalization.

## 6 CONCLUSIONS

In this paper, we discover the deep degradation representations in deep SR networks, which are different from high-level vision networks. We demonstrate that a well-trained deep SR network is naturally a good descriptor of degradation information. We reveal the differences in deep representations between classification and SR networks. We draw a series of interesting observations on the intrinsic features of deep SR networks, such as the effects of global residual and adversarial learning. Further, we apply DDR to several fundamental tasks and achieve appealing results. The exploration on DDR is of great significance and inspiration for relevant work.

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

# A APPENDIX

## A.1 BACKGROUND

Since the emergence of deep convolutional neural network (CNN), a large number of computer vision tasks have been drastically promoted, including high-level vision tasks such as image classification Russakovsky et al. (2015); Simonyan & Zisserman (2015); He et al. (2016); Huang et al. (2017); Hu et al. (2018), object localization Ren et al. (2015); He et al. (2017); Redmon et al. (2016) and semantic segmentation Long et al. (2015); Badrinarayanan et al. (2017); Chen et al. (2017); Wang et al. (2020a), as well as low-level vision tasks such as image super-resolution Dong et al. (2014); Ledig et al. (2017); Wang et al. (2018); Zhang et al. (2019); Dai et al. (2019), denoising Zhang et al. (2017; 2018a); Gu et al. (2019); Quan et al. (2020), dehazing Cai et al. (2016); Zhang & Patel (2018); Dong et al. (2020); Deng et al. (2020a), etc. However, an interesting phenomenon is that even if we have successfully applied CNNs to many tasks, yet we still do not have a thorough understanding of its intrinsic working mechanism.

To better understand the behaviors of CNN, many efforts have been put in the neural network interpretability for *high-level vision* Simonyan et al. (2013); Samek et al. (2017); Zeiler & Fergus (2014); Selvaraju et al. (2017); Montavon et al. (2018); Karpathy et al. (2015); Mahendran & Vedaldi (2016); Zhang et al. (2020); Adebayo et al. (2018). Most of them attempt to interpret the CNN decisions by visualization techniques, such as visualizing the intermediate feature maps (or saliency maps and class activation maps) Simonyan et al. (2013); Zeiler & Fergus (2014); Adebayo et al. (2018); Zhou et al. (2016); Selvaraju et al. (2017), computing the class notion images which maximize the class score Simonyan et al. (2013), or projecting feature representations Wen et al. (2016); Wang et al. (2020b); Zhu et al. (2018); Huang et al. (2020). For high-level vision tasks, especially image classification, researchers have established a set of techniques for interpreting deep models and have built up a preliminary understanding of CNN behaviors Gu et al. (2018). One representative work is done by Zeiler et al. Zeiler & Fergus (2014), who reveal the hierarchical nature of CNN by visualizing and interpreting the feature maps: the shallow layers respond to low-level features such as corners, curves and other edge/color conjunctions; the middle layers capture more complex texture combinations; the deeper layers are learned to encode more abstract and class-specific patterns, e.g., faces and legs. These patterns can be well interpreted by human perception and help partially explain the CNN decisions for high-level vision tasks.

As for *low-level vision* tasks, however, similar research work is absent. The possible reasons are as follows. In high-level vision tasks, there are usually artificially predefined semantic labels/categories. Thus, we can intuitively associate feature representations with these labels. Nevertheless, in low-level vision tasks, there is no explicit predefined semantics, making it hard to map the representations into a domain that the human can make sense of. Further, high-level vision usually performs *classification* in a discrete target domain with distinct categories, while low-level vision aims to solve a *regression* problem with continuous output values. Hence, without the guidance of predefined category semantics, it seems not so straightforward to interpret low-level vision networks.

In this paper, we take super-resolution (SR), one of the most representative tasks in low-level vision, as research object. Previously, it is generally thought that the features extracted from the SR network have no specific "semantic" information, and the network simply learns some complex non-linear functions to model the relations between network input and output. Are CNN features SR networks really in lack of any semantics? Can we find any kind of "semantics" in SR networks? In this paper, we aim to give an answer to these questions. We reveal that there **are** semantics existing in SR networks. We first discover and interpret the "semantics" of deep representations in SR networks. But different from high-level vision networks, such semantics relate to the image degradation types and degrees. Accordingly, we designate the deep semantic representations in SR networks as deep degradation representations (DDR).

## A.2 LIMITATIONS

In this paper, we only explore the deep representations of SR networks. Other low-level vision networks are also worth exploring. We apply DDR to three tasks without too elaborate design in the application parts. For blind SR, we make a simple attempt to improve the model performance. The design is not optimal. We believe that there should be a more efficient and effective way to utilize DDR. For generalization evaluation, DDR can only evaluate the model generalization under constrained conditions. It shows the possibility of designing a generalization evaluation metric, but there is still a long way to realize this goal.

## A.3 DEEP REPRESENTATIONS OF REAL-WORLD IMAGES

In the main paper, we mainly conduct experiments on synthetic degradations. The difficulty of real-world dataset is that it is hard to keep the content the same but change the degradations. If we simply use two real-world datasets which contains different contents and different degradations, it is hard to say whether the feature discriminability is targeted at image content or at image degradation. Hence, synthetic data at least can control the variables.

In addition, we find a plausible real-world dataset Real-City100, which is proposed in paper Cameral SR. The authors use iPhoneX and NikonD5500 devices to capture controllable images. By adjusting the cameral focal length, each camera captures paired images with the same content but different resolutions. The low-resolution images contain real-world degradations such as real noise and real

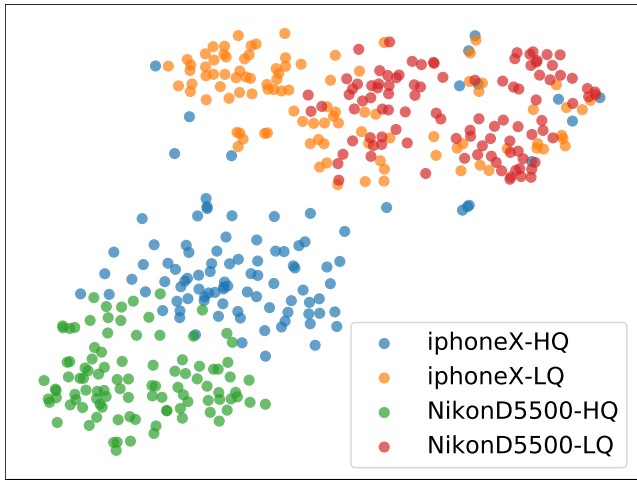

Figure 12: Projected feature representations of SRGAN-wGR on Real-City100 dataset.

blur. We test SRGAN on this dataset and obtain corresponding visualization results, as shown in 12. It can be seen that the deep representations of SRGAN can still distinguish among different degradations across different devices.

## A.4 CLASSIFICATION VS. SUPER-RESOLUTION

### A.4.1 FORMULATION

**Classification.** Classification aims to categorize an input image $X$ into a discrete object class:

$$\hat{Y} = G_{CL}(X), \tag{1}$$

where $G_{CL}$ represents the classification network, and $\hat{Y} \in \mathbb{R}^C$ is the predicted probability vector indicating which of the $C$ categories $X$ belongs to. In practice, cross-entropy loss is usually adopted to train the classification network:

$$CE(Y, \hat{Y}) = -\sum_{i=1}^{C} y_i log\hat{y}_i, \tag{2}$$

where $Y \in \mathbb{R}^C$ is a one-hot vector representing the ground-truth class label. $\hat{y}_i$ is the $i$-th row element of $\hat{Y}$, indicating the predicted probability that $X$ belongs to the $i$-th class.

**Super-resolution.** A general image degradation process can be model as follows:

$$X = (Y \otimes k) \downarrow_s +n, \tag{3}$$

where $Y$ is the high-resolution (HR) image and $\otimes$ denotes the convolution operation. $X$ is the degraded high-resolution (LR) image. There are three types of degradation in this model: blur kernel $k$, downsampling operation $\downarrow_s$ and additive noise $n$. Hence, super-resolution can be regarded as a superset of other restoration tasks like denoising and deblurring.

Super-resolution (SR) is the inverse problem of Equ. (3). Given the input LR image $X \in \mathbb{R}^{M \times N}$, the super-resolution network attempts to produce its HR version:

$$\hat{Y} = G_{SR}(X), \tag{4}$$

where $G_{SR}$ represents the super-resolution network, $\hat{Y} \in \mathbb{R}^{sM \times sN}$ is the predicted HR image and $s$ is the upscaling factor. This procedure can be regarded as a typical regression task. At present, there are two groups of method: **MSE-based** and **GAN-based** methods. The former one treats SR as a reconstruction problem, which utilizes pixel-wise loss such as $L_2$ loss to achieve high PSNR values.

$$L_2(Y, \hat{Y}) = \frac{1}{r^2 NM} \sum_{i=1}^{rN} \sum_{j=1}^{rM} \|Y_{i,j} - \hat{Y}_{i,j}\|_2^2. \tag{5}$$

This is the most widely used loss function in many image restoration tasks Dong et al. (2014); Lim et al. (2017); Zhang et al. (2018b;a); Cai et al. (2016); He et al. (2020). However, such loss tends to produce over-smoothed images. To generate photo-realistic SR results, the latter method incorporates adversarial learning and perceptual loss to benefit better visual perception. The optimization is expressed as following min-max problem:

$$\min_{\theta_{G_{SR}}} \max_{\theta_{D_{SR}}} \mathbb{E}_{Y \sim p_{HR}}[\log D_{SR}(Y)]$$
$$+ \mathbb{E}_{X \sim p_{LR}}[\log(1 - D_{SR}(G_{SR}(X)))]. \tag{6}$$

In such adversarial learning, a discriminator $D_{SR}$ is introduced to distinguish super-resolved images from real HR images. Then, the generator loss is defined as:

$$L_G = -\log D_{SR}(G_{SR}(X)). \tag{7}$$

From the formulation, we can clearly see that image classification and image super-resolution represent two typical tasks in machine learning: classification and regression. The output of the classification task is discrete, while the output of the regression task is continuous.

### A.4.2 ARCHITECTURES

Due to the different output types, the CNN architectures of classification and super-resolution networks also differ. Generally, classification networks often contain multiple downsampling layers (e.g., pooling and strided convolution) to gradually reduce the spatial resolution of feature maps. After several convolutional and downsampling layers, there may be one or more fully-connected layers to aggregate global semantic information and generate a vector containing $C$ elements. For the output layer, the SoftMax operator is frequently used to normalize the previously obtained vector into a probabilistic representation. Some renowned classification network structures include AlexNet Krizhevsky et al. (2012), VGG Simonyan & Zisserman (2015), ResNet He et al. (2016), InceptionNet Szegedy et al. (2015); Ioffe & Szegedy (2015); Szegedy et al. (2017), DenseNet Huang et al. (2017), SENetBadrinarayanan et al. (2017), etc.

Unlike classification networks, super-resolution networks usually do not rely on downsampling layers, but upsampling layers (e.g., bilinear upsampling, transposed convolution Zeiler et al. (2010) or subpixel convolution Shi et al. (2016)). Thus, the spatial resolution of feature maps would increase. Another difference is that the output of the SR network is a three-channel image, rather than an abstract probability vector. The well-known SR network structures include SRCNN Dong et al. (2014), FSRCNN Dong et al. (2016), SRResNet Ledig et al. (2017), RDN Zhang et al. (2018c), RCAN Zhang et al. (2018b), etc. An intuitive comparison of classification and SR networks in CNN architecture is shown in Fig. 18. We can notice that one is gradually downsampling, and the other is gradually upsampling, which displays the discrepancy between high-level vision and low-level vision tasks in structure designing.

Although there are several important architectural differences, classification networks and SR networks can share and adopt some proven effective building modules, like skip connection He et al. (2016); Lim et al. (2017) and attention mechanismHu et al. (2018); Zhang et al. (2018b).

### A.5 IMPLEMENTATION DETAILS

In the main paper, we conduct experiments on ResNet18 He et al. (2016) and SRResNet/SRGAN Ledig et al. (2017). We elaborate more details on the network structures and training settings here.

For ResNet18, we directly adopt the network structure depicted in He et al. (2016). Cross-entropy loss (Eq. 2) is used as the loss function. The learning rate is initialized to $0.1$ and decreased with a cosine annealing strategy. We apply SGD optimizer with weight decay $5 \times 10^{-4}$. The trained model yields an accuracy of $92.86\%$ on CIFAR10 testing set which consists of $10,000$ images.

For SRResNet-wGR/SRResNet-woGR, we stack 16 residual blocks (RB) as shown in Fig. 3 of the main paper. The residual block is the same as depicted in Wang et al. (2018), in which all the BN layers are removed. Two Pixel-shuffle layers Shi et al. (2016) are utilized to conduct upsampling in the network, while the global residual branch is upsampled by bilinear interpolation. $L_1$ loss is adopted as the loss function. The learning rate is initialized to $2 \times 10^{-4}$ and is halved at $[100k, 300k, 500k, 600k]$ iterations. A total of $600,000$ iterations are executed.

For SRGAN-wGR/SRGAN-woGR, the generator is the same as SRResNet-wGR/SRResNet-woGR. The discriminator is designed as in Ledig et al. (2017). Adversarial loss (Eq. 7) and perceptual loss Johnson et al. (2016) are combined as the loss functions, which are kept the same as in Ledig et al. (2017). The learning rate of both generator and discriminator is initialized to $1 \times 10^{-4}$ and is halved at $[50k, 100k, 200k, 300k]$ iterations. A total of $600,000$ iterations are executed. For all the super-resolution networks, we apply Adam optimizer Kingma & Ba (2014) with $\beta_1 = 0.9$ and $\beta_2 = 0.99$. All the training LR patches are of size $128 \times 128$. When testing, $32 \times 32$ patches are fed into the networks to obtain deep features. In practice, we find that the patch size has little effect on revealing the deep degradation representations. All above models are trained on PyTorch platform with GeForce RTX 2080 Ti GPUs.

For the experiment of distortion identification, we use the aforementioned trained models to conduct inferencing on the LIVE dataset Sheikh et al. (2006). We crop the central $96 \times 96$ patch of each image to feed into the SR networks and obtain the corresponding deep representations. Then, the deep representations of each image are reduced to 120-dimensional vector using PCA. Afterwards, the linear SVM is adopted as the classification tail. In practice, we find that the vector dimension can be even larger for better performance. Notably, unlike previous methods, the features here are not trained on any degradation related labels or signals. The SR networks are only trained using clean data. However, the deep representations can be excellent prior features for recognizing various distortion types. This is of great importance and very encouraging.

### A.6 Definitions of WD, BD and CHI

In Sec. 3.1 of the main paper, we describe the adopted analysis method on deep feature representations. Many other literatures also have adopted similar approaches to interpret and visualize the deep models, such as Graph Attention Network Veličković et al. (2017), Recurrent Networks Karpathy et al. (2015), Deep Q-Network Zahavy et al. (2016) and Neural Models in NLP Li et al. (2015). Most aforementioned researches adopt t-SNE as a qualitative analysis technique. To better illustrate and quantitatively measure the semantic discriminability of deep feature representations, we take a step further and introduce several indicators, which are originally used to evaluate the clustering performance, according to the data structure after dimensionality reduction by t-SNE. Specifically, we propose to adopt within-cluster dispersion (WD), between-clusters dispersion (BD) and Calinski-Harabaz Index (CHI) Caliński & Harabasz (1974) to provide some rough yet practicable quantitative measures for reference. For $K$ clusters, WD, BD and CHI are defined as:

$$WD(K) = \sum_{k=1}^{K} \sum_{i=1}^{n(k)} \|\boldsymbol{x}_k^i - \bar{\boldsymbol{x}}_k\|^2, \tag{8}$$

where $\boldsymbol{x}_k^i$ represents the $i$-th datapoint belonging to class $k$ and $\bar{\boldsymbol{x}}_k$ is the average mean of all $n(k)$ datapoints that belong to class $k$. Datapoints belonging to the same class should be close enough to each other and WD measures the compactness within a cluster.

$$BD(K) = \sum_{k=1}^{K} n(k) \|\bar{\boldsymbol{x}}_k - \bar{\boldsymbol{x}}\|^2, \tag{9}$$

where $\bar{\boldsymbol{x}}$ represents the average mean of all datapoints. BD measures the distance between clusters. Intuitively, larger BD value indicates stronger discriminability between different feature clusters. Given $K$ clusters and $N$ datapoints in total ($N = \sum_k n(k)$), by combining WD and BD, the CHI is formulated as:

$$CHI(K) = \frac{BD(K)}{WD(K)} \cdot \frac{(N - K)}{(K - 1)}. \tag{10}$$

It is represented as the ratio of the between-clusters dispersion mean and the within-cluster dispersion. The CHI score is higher when clusters are dense and well separated, which relates to a standard concept of a cluster.

**Rationality of Using Quantitative Measures with t-SNE.** Notably, t-SNE is not a numerical technique but a probabilistic one. It minimizes the Kullback-Leibler (KL) divergence between the dis-

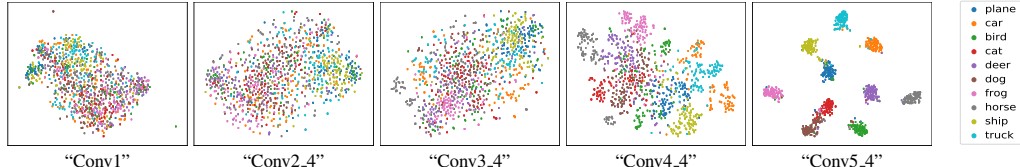

| "Conv1" | "Conv2_4" | "Conv3_4" | "Conv4_4" | "Conv5_4" |

Figure 13: Projected feature representations extracted from different layers of ResNet18 using t-SNE. With the network deepens, the representations become more discriminative to object categories, which clearly shows the semantics of the representations in classification.

| #Layer | Conv1 | Conv2_4 | Conv3_4 | Conv4_4 | Conv5_4 |
|---|---|---|---|---|---|
| Dim | 64×32×32 | 64×32×32 | 128×16×16 | 256×8×8 | 512×4×4 |
| WD ↓ ($\times10^5$) | 4.07 ± 0.43 | 3.41 ± 0.31 | 3.32 ± 0.31 | 2.06 ± 0.13 | 0.71 ± 0.06 |
| BD ↑ ($\times10^5$) | 1.04 ± 0.13 | 1.22 ± 0.11 | 1.84 ± 0.40 | 5.77 ± 0.23 | 10.74 ± 0.20 |
| CHI ↑ | 28.18 ± 1.69 | 39.22 ± 1.44 | 61.12 ± 13.62 | 309.31 ± 31.10 | 1688.62 ± 145.15 |

Table 3: Quantitative measures for the discriminability of the projected deep feature representations. We statistically report the mean value and the standard deviation of each metric. The adopted indicators well reflect the effect of feature clustering quantitatively.

| | Set5 | Set14 | Urban100 | DIV2K |
|---|---|---|---|---|
| SRCNN-3L | 28.51 | 25.72 | 22.86 | 27.80 |
| SRCNN-5L | 28.89 | 25.99 | 23.22 | 28.05 |
| SRCNN-7L | 28.97 | 26.02 | 23.27 | 28.09 |
| SRCNN-9L | 29.17 | 26.17 | 23.48 | 28.24 |
| SRCNN-11L | 29.27 | 26.21 | 23.56 | 28.29 |
| SRCNN-13L | 29.39 | 26.28 | 23.66 | 28.36 |

Table 4: The PSNR values of SRCNN with different depth on classical SR benchmark datasets.

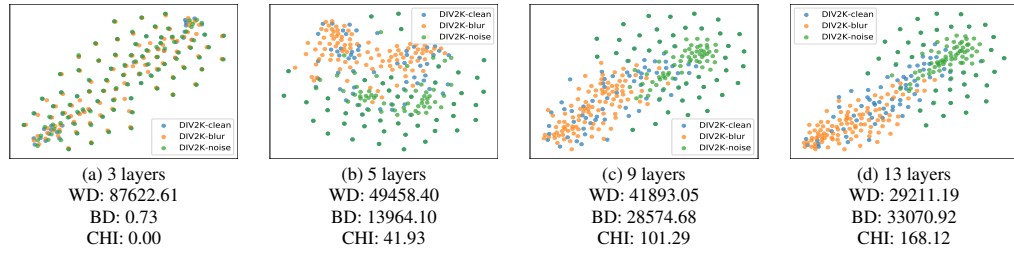

| (a) 3 layers | (b) 5 layers | (c) 9 layers | (d) 13 layers |
|---|---|---|---|
| WD: 87622.61 | WD: 49458.40 | WD: 41893.05 | WD: 29211.19 |
| BD: 0.73 | BD: 13964.10 | BD: 28574.68 | BD: 33070.92 |
| CHI: 0.00 | CHI: 41.93 | CHI: 101.29 | CHI: 168.12 |

Figure 14: With more layers, the model deep representations gradually manifest the discriminability on degradation types.

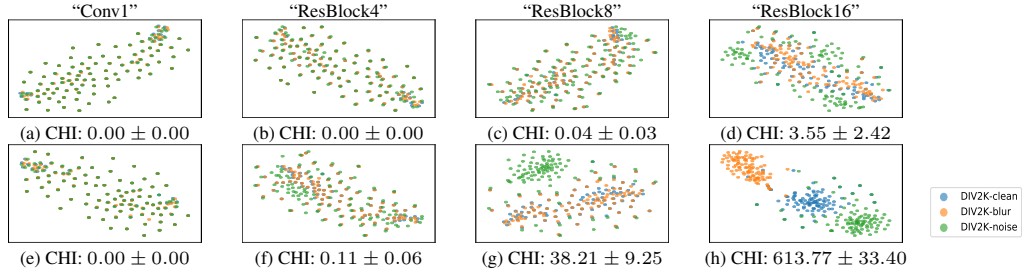

Figure 15: Projected feature representations extracted from different layers of SRResNet-woGR (1st row) and SRResNet-wGR (2nd row) using t-SNE. With image global residual (GR), the representations of MSE-based SR networks show discriminability to degradation types.

tributions that measure pairwise similarities of the input high-dimensional data and that of the corresponding low-dimensional points in the embedding. Further, t-SNE is a non-convex optimization process which is performed using a gradient descent method, as a result of which several optimization parameters need to be chosen, like perplexity, iterations and learning rate. Hence, the reconstruction solutions may differ due to the choice of different optimization parameters and the initial random states. In this paper, we used exactly the same optimization procedure for all experiments. Moreover, we conduct extensive experiments using different parameters and demonstrate that the quality of the optima does not vary much from run to run, which is also emphasized in the t-SNE paper. To make the quantitative analysis more statistically solid, for each projection process, we run t-SNE five times and report the average and standard deviations of every metric.

## A.7 FROM SHALLOW TO DEEP SR NETWORKS

In the main paper, we reveal that a shallow 3-layer SRCNN Dong et al. (2014) does not manifest representational discriminability on degradation types. Thus, we hypothesize that only deep SR networks possess such degradation-related semantics. To verify the statement, we gradually deepen the depth of SRCNN and observe how its deep representations change. We construct SRCNN models with different layer depths from shallow 3 layers to 13 layers. We train these models on DIV2K-clean data (inputs are only downsampled without other degradations) and test them on classical SR benchmarks. As shown in Tab. 4, the model achieves better SR performance with the increase of network depth, suggesting that deeper networks and more parameters can lead to greater learning capacity. On the other hand, the deep representations also gradually manifest discriminability on degradation types, as depicted in Fig. 14. When the model only has 3 layers, its representations cannot distinguish different degradation types. However, when we increase the depth to 13 layers, the deep representations begin to show discriminability on degradation types, with the CHI score increasing to 168.12.

| SRResNet-woGR | | | | |
|---|---|---|---|---|
| #Layer | Conv1 | ResBlock4 | ResBlock8 | ResBlock16 |
| WD$\downarrow$($\times 10^4$) | $8.35 \pm 0.14$ | $8.90 \pm 0.22$ | $9.28 \pm 0.31$ | $4.98 \pm 0.48$ |
| BD$\uparrow$ | $0.29 \pm 0.14$ | $1.98 \pm 1.47$ | $25.60 \pm 17.73$ | $1149.20 \pm 765.12$ |
| CHI$\uparrow$ | $0.00 \pm 0.00$ | $0.00 \pm 0.00$ | $0.04 \pm 0.03$ | $3.55 \pm 2.42$ |
| SRResNet-wGR | | | | |
| #Layer | Conv1 | ResBlock4 | ResBlock8 | ResBlock16 |
| WD$\downarrow$($\times 10^4$) | $8.20 \pm 0.18$ | $8.40 \pm 0.09$ | $4.40 \pm 0.50$ | $0.86 \pm 0.11$ |
| BD$\uparrow$ | $0.48 \pm 0.34$ | $62.74 \pm 33.99$ | $11096.79 \pm 2051.02$ | $35470.66 \pm 4412.66$ |
| CHI$\uparrow$ | $0.00 \pm 0.00$ | $0.11 \pm 0.06$ | $38.21 \pm 9.25$ | $613.77 \pm 33.40$ |

Table 5: Quantitative measures for the projected deep feature representations obtained by SRResNet-woGR and SRResNet-wGR.

## A.8 MORE APPLICATIONS

**Evaluating the Generalization Ability.** According to the discussions in Sec. 4.6, DDR can be used as an approximate evaluation metric for generalization ability. Specifically, given a trained model and several test datasets with different degradations, we can obtain their DDR features. By

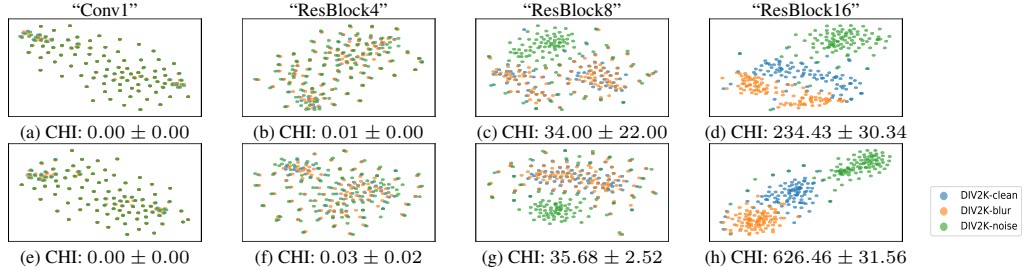

Figure 16: Projected feature representations extracted from different layers of SRGAN-woGR (1st row) and SRGAN-wGR (2nd row) using t-SNE. Even without GR, GAN-based SR networks can still obtain deep degradation representations.

| SRGAN-woGR | | | | |
|---|---|---|---|---|
| #Layer | Conv1 | ResBlock4 | ResBlock8 | ResBlock16 |
| WD↓$(\times 10^4)$ | $7.94 \pm 0.20$ | $7.83 \pm 0.33$ | $4.65 \pm 0.58$ | $1.44 \pm 0.28$ |
| BD↑ | $0.58 \pm 0.41$ | $4.79 \pm 2.43$ | $9809.00 \pm 4501.19$ | $22459.35 \pm 3560.33$ |
| CHI↑ | $0.00 \pm 0.00$ | $0.01 \pm 0.00$ | $34.00 \pm 22.00$ | $234.43 \pm 30.34$ |
| SRGAN-wGR | | | | |
| #Layer | Conv1 | ResBlock4 | ResBlock8 | ResBlock16 |
| WD↓$(\times 10^4)$ | $7.47 \pm 0.20$ | $7.97 \pm 0.19$ | $4.83 \pm 0.52$ | $0.72 \pm 0.10$ |
| BD↑ | $0.41 \pm 0.36$ | $14.89 \pm 8.85$ | $11600.91 \pm 1424.10$ | $30180.52 \pm 2884.65$ |
| CHI↑ | $0.00 \pm 0.00$ | $0.03 \pm 0.02$ | $35.68 \pm 2.52$ | $626.46 \pm 31.56$ |

Table 6: Quantitative measures for the projected deep feature representations obtained by SRGAN-woGR and SRGAN-wGR.

evaluating the discriminability of the projection results (clustering effect), we can roughly measure the generalization performance over different degradation types. The worse the clustering effect, the better the generalizability. Fig .11 shows the DDR clustering of different models. RRDB (clean) is unable to deal with degraded data and obtains lower PSNR values on blur and noise inputs. Its CHI score is 322.16. By introducing degraded data into training, the model gains better generalization and the CHI score is 14.04. With DDR guidance, the generalization ability is further enhanced. The CHI score decreases to 4.95. The results are consistent with the results in the previous section. Interestingly, we do not need ground-truth images to evaluate the model generalization. A similar attempt has been made in recent work Liu et al. (2022). Note that CHI is only a rough index, which cannot accurately measure the minor differences. DDR shows the possibility of designing a generalization evaluation metric, but there is still a long way to realize this goal.

## A.9 EXPLORATION ON DIFFERENT DEGRADATION DEGREES

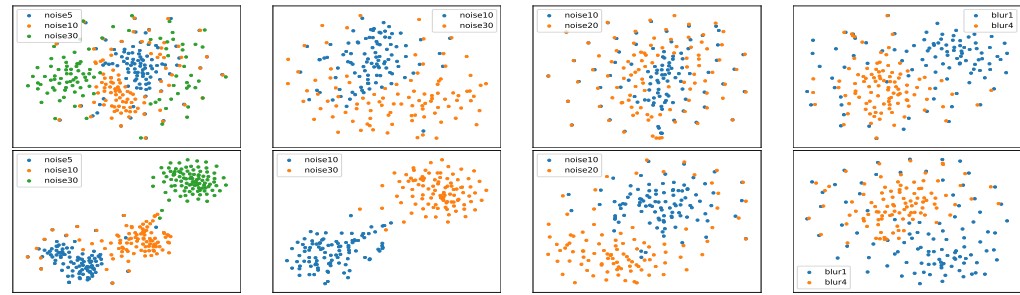

Figure 17: Even for the same type of degradation, different degradation degrees will also cause differences in features. The greater the difference between degradation degrees, the stronger the discriminability. **First row:** SRResNet-wGR. **Second row:** SRGAN-wGR.

Previously, we introduce deep degradation representations by showing that the deep representations of SR networks are discriminative to different degradation types (e.g., clean, blur and noise). How about the same degradation type but with different degraded degrees? Will the deep representa-

| | | Cross-degradation | Intra-degradation (degradation degrees) | | | |
|---|---|---|---|---|---|---|
| | structure | Clean-Blur-Noise | Noise{5,10,30} | Noise{10,30} | Noise{10,20} | Blur{1,4} |
| SRResNet | woGR | - (3.55) | - (6.29) | - (7.84) | - (0.23) | - (0.02) |
| | GR | +++ (613.77) | - (36.53) | + (41.50) | - (0.59) | + (53.37) |
| MSRGAN | woGR | ++ (234.43) | +++ (551.26) | +++ (525.55) | + (52.67) | - (1.40) |
| | wGR | +++ (626.46) | +++ (815.11) | +++ (831.35) | + (79.40) | + (35.04) |

\* -: $0 \sim 20$. +: $20 \sim 100$. ++: $100 \sim 500$. +++: $\geq 500$.

Table 7: Quantitative evaluations (CHI). There appears to be a spectrum (continuous transition) for the discriminability of DDR.

tions still be discriminative to them? To explore this question, more experiments and analysis are performed.

We test super-resolution networks on degraded images with different noise degrees and blur degrees. The results are depicted in Table. 7 and Fig. 17. It can be seen that the deep degradation representations are discriminative not only to cross-degradation (different degradation types) but also to intra-degradation (same degradation type but with different degrees). This suggests that even for the same type of degradation, different degradation degrees will also cause significant differences in features. The greater the difference between degradation degrees, the stronger the discriminability of feature representations. This also reflects another difference between the representation semantics of super-resolution network and classification network. For classification, the semantic discriminability of feature representations is generally discrete, because the semantics are associated with discrete object categories. Nevertheless, there appears to be a spectrum (continuous transition) for the discriminability of the deep degradation representations, i.e., the discriminability has a monotonic relationship with the divergence between degradation types and degrees. For example, the degradation difference between noise levels 10 and 20 is not that much distinct, and the discriminability of feature representations is relatively smaller, comparing with noise levels 10 and 30.

From Table 7, there are notable observations. 1) Comparing with blur degradation, noise degradation is easier to be discriminated. Yet, it is difficult to obtain deep representations that have strong discriminability for different blur levels. Even for GAN-based method, global residual (GR) is indispensable to obtain representations that can be discriminative to different blur levels. 2) The representations obtained by GAN-based method have more discriminative semantics to degradation types and degrees than those of MSE-based method. 3) Again, global residual can strengthen the representation discriminability for degradations.

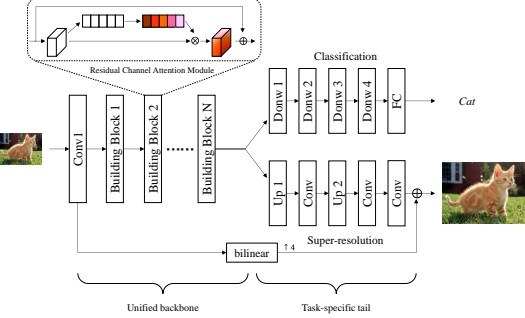

Figure 18: Unified backbone framework for classification and super-resolution. The two networks share the same backbone structure and different tails.

## A.10 EXPLORATION OF NETWORK STRUCTURE

In the main paper, we choose ResNet18 He et al. (2016) and SRResNet/SRGAN Ledig et al. (2017) as the backbones of classification and SR networks, respectively. In order to eliminate the influence of different network structures, we design a unified backbone framework, which is composed of the

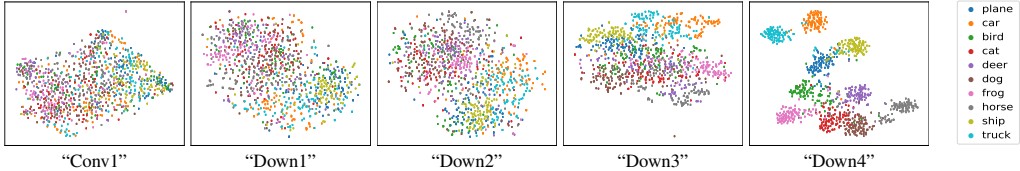

"Conv1"   "Down1"   "Down2"   "Down3"   "Down4"

Figure 19: Projected feature representations extracted from different layers of unified backbone framework (classification) using t-SNE. The results are similar to ResNet18, which validates that the deep semantic representations are uncorrelated with network structures but are associated with the task itself.

| #Layer | Conv1 | Down1 | Down2 | Down3 | Down4 |
|---|---|---|---|---|---|
| Dim | 64×32×32 | 64×32×32 | 128×16×16 | 256×8×8 | 512×4×4 |
| WD ↓ ($\times 10^5$) | 3.64 ± 0.33 | 2.76 ± 0.27 | 2.52 ± 0.19 | 1.83 ± 0.05 | 0.59 ± 0.02 |
| BD ↑ ($\times 10^5$) | 1.10 ± 0.13 | 0.97 ± 0.18 | 1.60 ± 0.19 | 3.84 ± 0.40 | 7.48 ± 0.32 |
| CHI ↑ | 33.11 ± 1.38 | 39.53 ± 9.98 | 70.11 ± 9.94 | 230.95 ± 22.63 | 1403.96 ± 27.17 |

Table 8: Quantitative measures for the discriminability of the projected deep feature representations obtained by unified backbone framework (classification).

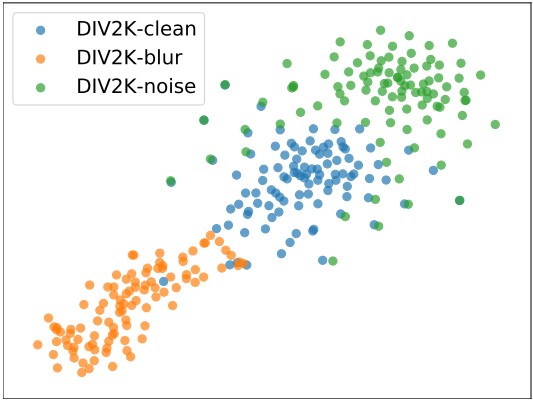

Figure 20: Projected feature representations extracted from unified backbone framework (super-resolution) using t-SNE.

same basic building modules but connected with different tails for downsampling and upsampling to conduct classification and super-resolution respectively.

The unified architecture is shown in Fig. 18. To differ from the residual block in the main paper, we adopt residual channel attention layer as basic building block, which is inspired by SENet Hu et al. (2018) and RCAN Zhang et al. (2018b). For classification, the network tail consists of three maxpooling layers and a fully connected layer; for super-resolution, the network tail consists of two pixel-shuffle layers to upsample the feature maps. According to the conclusions in the main paper, we adopt global residual (GR) in the network design to obtain deep degradation representations (DDR). Except the network structure, all the training protocols are kept the same as in the main paper. The training details are the same as depicted in Sec. A.5. After training, the unified backbone framework for classification yields an accuracy of $92.08\%$ on CIFAR10 testing set.

The experimental results are shown in Fig. 19, Fig. 20 and Tab. 8. From the results, we can see that the observations are consistent with the findings in the main paper. It suggests that the semantic representations do not stem from network structures, but from the task itself. Hence, our findings are not only limited to specific structures but are universal.

### A.11 MORE INSPIRATIONS AND FUTURE WORK

**Disentanglement of Image Content and Degradation** In plenty of image editing and synthesizing tasks, researchers seek to disentangle an image through different attributes, so that the image can be finely edited Karras et al. (2019); Ma et al. (2018); Deng et al. (2020b); Lee et al. (2018); Nitzan et al. (2020). For example, semantic face editing Shen et al. (2020a;b); Shen & Zhou (2020) aims at manipulating facial attributes of a given image, e.g., pose, gender, age, smile, etc. Most methods attempt to learn disentangled representations and to control the facial attributes by manipulating the latent space. In low-level vision, the deep degradation representations can make it possible to decompose an image into content and degradation information, which can promote a number of new areas, such as degradation transferring and degradation editing. Further, more in-depth research on deep degradation representations will also greatly improve our understanding of the nature of images.

### A.12 DISCUSSIONS ON DIMENSIONALITY REDUCTION

Among the numerous dimensionality reduction techniques (e.g., PCA Hotelling (1933), CCA Demartines & Hérault (1997), LLE Roweis & Saul (2000), IsomapTenenbaum et al. (2000), SNEHinton & Roweis (2002)), t-Distributed Stochastic Neighbor Embedding (t-SNE) Van der Maaten & Hinton (2008) is a widely-used and effective algorithm. It can greatly capture the local structure of the high-dimensional data and simultaneously reveal global structure such as the presence of clusters at several scales. Following Donahue et al. (2014); Mnih et al. (2015); Wen et al. (2016); Zahavy et al. (2016); Veličković et al. (2017); Wang et al. (2020b); Huang et al. (2020), we also take advantage of the superior manifold learning capability of t-SNE for feature projection.

In this section we further explain the effectiveness of adopting t-SNE and why we choose to project hign-dimensional features into two-dimensional datapoints. We first compare the projection results of PCA and t-SNE. From the results shown in Fig. 21, it can be observed that the projected features by t-SNE are successfully clustered together according the semantic labels, while the projected features by PCA are not well separated. It is because that PCA is a linear dimension reduction method which cannot deal with complex non-linear data obtained by the neural networks. Thus, t-SNE is a better choice to conduct dimension reduction on CNN features. This suggests the effectiveness of t-SNE for the purpose of feature projection. Note that we do not claim t-SNE is the optimal or the best choice for dimensionality reduction. We just utilize t-SNE as a rational tool to show the trend behind deep representations, since t-SNE has been proven effective and practical in our experiments and other literatures.

Then, we discuss the dimensions to reduce. We conduct dimensionality reduction to different dimensions. Since the highest dimension supported by t-SNE is 3, we first compare the effect between the two-dimensional projected features and the three-dimensional projected features by t-SNE. The qualitative and quantitative results are shown in Fig. 21 and Tab. 9. When we reduce the features to three dimensions, the reduced representations also show discriminability to semantic labels. How-

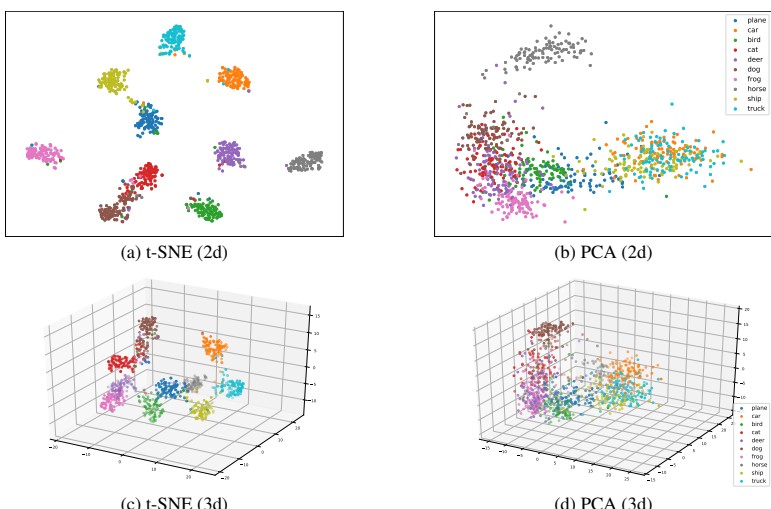

(a) t-SNE (2d)  (b) PCA (2d)

(c) t-SNE (3d)  (d) PCA (3d)

Figure 21: Comparison between PCA and t-SNE for projecting feature representations ("Conv5_4" layer of ResNet18).

ever, quantitative results show that two dimensions can better portray the discriminability than three or higher dimensions. For PCA, the results are similar. With higher dimensions, the discriminability decrease. Hence, it is reasonable to reduce high-dimensional features into two-dimensional data-points. Such settings are also adopted in Donahue et al. (2014); Wang et al. (2020b); Veličković et al. (2017); Huang et al. (2020), which are proven effective.

| #Layer | Conv5_4 | | | | | |
|---|---|---|---|---|---|---|
| Input #Dim | $512 \times 4 \times 4$ | | | | | |
| Method | PCA(50)+t-SNE(2) | PCA(50)+t-SNE(3) | PCA | PCA | PCA | PCA |
| Reduced #Dim | 2 | 3 | 2 | 3 | 4 | 5 |
| WD $\downarrow$ ($\times 10^5$) | $0.71 \pm 0.06$ | $0.24 \pm 0.06$ | 0.19 | 0.32 | 0.39 | 0.47 |
| BD $\uparrow$ ($\times 10^5$) | $10.74 \pm 0.20$ | $2.09 \pm 0.04$ | 1.27 | 1.61 | 1.95 | 2.24 |
| CHI $\uparrow$ | $1688.62 \pm 145.15$ | $978.58 \pm 224.77$ | 729.64 | 562.85 | 554.92 | 526.64 |

Table 9: Quantitative comparison with dimensionality reduction methods and reduced dimensions. To utilize t-SNE, we first use PCA to pre-reduce the features to 50 dimensions. Since PCA is a numerical method, the result is fixed. For t-SNE, we report the mean and standard deviation for 5 runs. The quantitative results show that t-SNE surpasses PCA and reducing to two dimensions is better. The features are obtained by "Conv5_4" layer of ResNet18.

## A.13 VISUALIZATION OF FEATURE MAPS

So far, we have successfully revealed the degradation-related semantics in SR networks with dimensionality reduction. In this section, we directly visualize the deep feature maps extracted from SR networks to provide some intuitive and qualitative interpretations. Specifically, we extract the feature maps obtained from four models (SRResNet-wGR, SRResNet-woGR, SRGAN-wGR and SRGAN-woGR) on images with different degradations (clean, blur4, noise20), respectively. Then we treat each feature map as a one channel image and plot it. The visualized feature maps are shown in Fig. 22. We select 8 feature maps with the largest eigenvalues for display. The complete results are shown in the supplementary file.

**Influence of degradations on feature maps.** From Fig. 22(a), we can observe that the deep features obtained by SRResNet-woGR portray various characteristics of the input image, including edges, textures and contents. In particular, we highlight in "red rectangles" the features that retain most of the image content. As shown in Fig. 22(b), after applying blur and noise degradations to the input image, the extracted features appear similar degradations as well. For blurred/noisy input images, the extracted feature maps also contain homologous blur/noise degradations.

**Effect of global residual.** In Sec. 4.3, we have revealed the importance and effectiveness of global residual (GR) for obtaining deep degradation representations for SR networks. But why GR is so

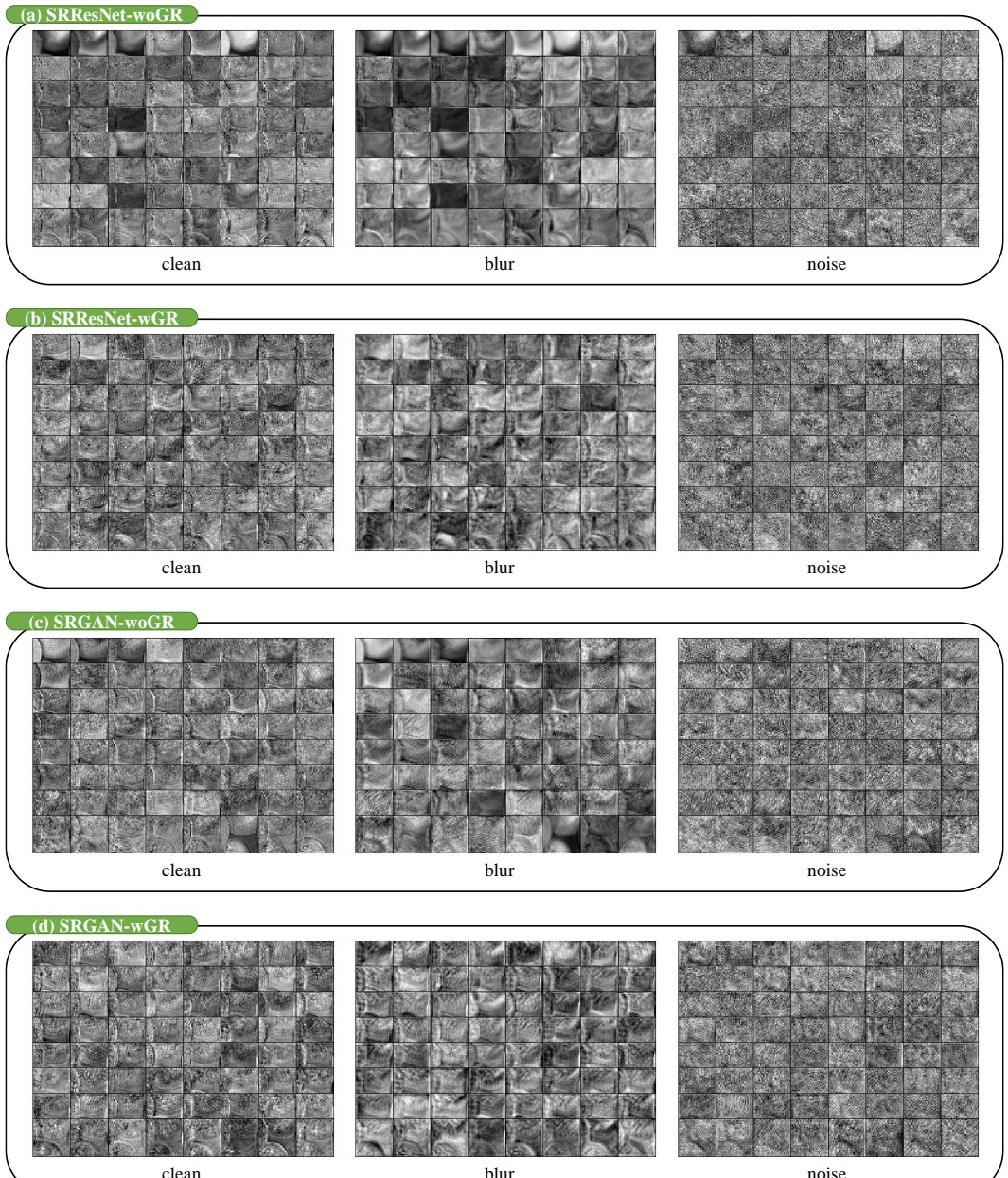

Figure 22: Visualization of feature maps. GR and GAN can facilitate the network to obtain more features on degradation information.

important? What is the role of GR? Through visualization, we can provide a qualitative and intuitive explanation here. Comparing Fig. 22(a) and Fig. 22(b), it can be observed that by adopting GR, the extracted features seem to contain less components of original shape and content information. Thus, GR can help remove the redundant image content information and make the network concentrate more on obtaining features that are related to low-level degradation information.

**Effect of GAN.** Previously, we have discussed the difference between MSE-based and GAN-based SR methods in their deep representations. We find that GAN-based method can better obtain feature representations that are discriminative to different degradation types. As shown in Fig. 22(a) and Fig. 22(c), the feature maps extracted by GAN-based method contain less object shape and content information compared with MSE-based method. This partially explains why the deep representations of GAN-based method are more discriminative, even without global residual. Comparing Fig. 22(c) and Fig. 22(d), when there is global residual, the feature maps containing the image original content information are further reduced, leading to stronger discriminability to degradation types.

## A.14 SAMPLES OF DIFFERENT DATASETS

In the main paper, we adopt several different datasets to conduct experiments. Fig. 23 displays some example images from these datasets.

(a) DIV2K-clean: the original DIV2K Agustsson & Timofte (2017) dataset. The high-resolution (HR) ground-truth (GT) images have 2K resolution and are of high visual quality. The low-resolution (LR) input images are downsampled from HR by bicubic interpolation, without any further degradations.

(b) DIV2K-noise: adding Gaussian noises to DIV2K-clean LR input, thus making it contain extra noise degradation. DIV2K-noise20 means the additive Gaussian noise level $\sigma$ is 20, where the number denotes the noise level.

(c) DIV2K-blur: applying Gaussian blur to DIV2K-clean LR input, thus making it contain extra blur degradation. DIV2K-blur4 means the Gaussian blur width is 4.

(d) DIV2K-mild: officially synthesized from DIV2K Agustsson & Timofte (2017) dataset as challenge dataset Timofte et al. (2017; 2018), which contains noise, blur, pixel shifting and other degradations. The degradation modelling is unknown to challenge participants.

(e) Hollywood100: 100 images selected from Hollywood dataset Laptev et al. (2008), containing real-world old film frames with unknown degradations, which may have compression, noise, blur and other real-world degradations.

Dataset (a), (b), (c) and (d) have the same image contents but different degradations. However, we find that the deep degradation representations (DDR) obtained by SR networks have discriminability to these degradation types, even if the network has not seen these degradations at all during training. Further, for real-world degradation like in (e), the DDR are still able to discern it.

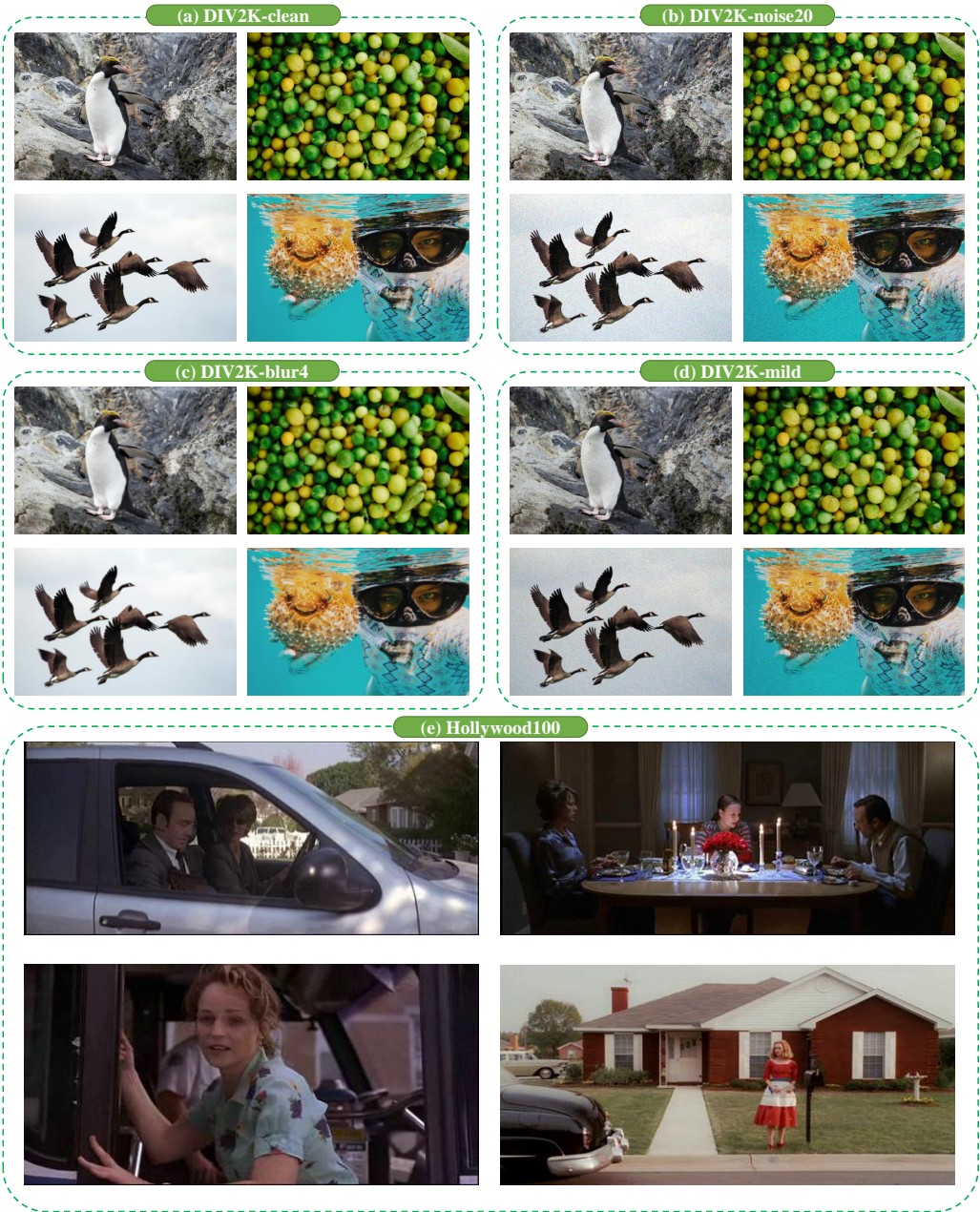

Figure 23: Example images from different datasets. (a) DIV2k-clean. (b) DIV2k-noise20. (c) DIV2k-blur4. (d) DIV2k-mild. (e) Hollywood100. Different datasets contain different degradation types. (a), (b), (c) and (d) are aligned with image content, but contains degradations. The deep degradation representations (DDR) are discriminative to various degradations.

