# OpenReview forum: "Discovering Distinctive ``Semantics'' in Super-Resolution Networks"
_ICLR.cc/2023/Conference — Submitted to ICLR 2023_

### Official Review · Reviewer_xxDc · 2022-10-24

**Confidence:** 3
**Correctness:** 1
**Technical Novelty And Significance:** 2
**Empirical Novelty And Significance:** 2
**Recommendation:** 5

**Clarity, Quality, Novelty And Reproducibility:**

- This paper is quite clear and has a novel point.

- Detailed Implementation information is somewhat lacking, but overall it is judged that it can be reproduced.

- It has not been verified whether the performance of the real SOTA blind SR model can be improved or whether there are other practical applications. Experiments on distortion identification and blind SR presented in the paper only show that it is applicable to them. It should be demonstrated experimentally whether it can effectively improve the existing distortion identification and blind SR methods.


**Strength And Weaknesses:**

Strength
- In this paper, interesting findings are introduced such as degradation-related semantics.
- The proposed findings can be adopted for various low-level vision tasks.
- It is interesting that we need to diminish the feature discriminability to improve the model generalization for various degradations.

Weaknesses
- Although the observation and finding of this paper are interesting, the finding of this paper should be verified by real applications. Although applied to distortion identification and blind SR, it is somewhat insufficient. Comparative analysis with existing blind SR techniques such as IKC is required, and experiments on other degradations such as compressing rather than noise and blur are also required.

- I think that this paper mainly focuses on the SR. However, in the motivation section, experiments for observations are biased to image denoising (or restoration). Also, compared to the traditional method is BM3D which is a representative denoising method. When the reader first sees the paper, it seems strange. Figure 2 should be an experiment on SR.

- Figure 2 does not show observations for various degradations. Not only noise, but also degradation such as compression, irregular holes, and color tones should be dealt with as well.

- Figure 5~8 seems to be convincing only if more diverse degradations. In addition, it is necessary to verify whether there is a different phenomenon that varies depending on the upsampling factor of the SR network.

- Minor points
    - [1pp] sparse coding(Yang ) → sparse coding (Yang )
    - [3pp] Wang et al. Wang et al. (2020b) → Wang et al. (Wang et al. (2020b))
    - In Figure 2, the results in the last column are from SRCNN? Text above the figure denotes it as SRCNN but texts in the caption denotes as CinCGAN (Yuan et al., 2018) is trained on DIV2K-mild dataset in an unpaired manner.


**Summary Of The Paper:**

This paper analyzes semantics learned in the SR network, and finds that a well trained SR network can naturally extract degradation descriptors. Authors find that conventional or shallow CNN models have difficulty in separating degradation-related semantics. Especially, global residual and adversarial learnings make the SR network extract degradation-related representation. Then, authors exploit their findings for other vision tasks.

**Summary Of The Review:**

This paper provides interesting views in the SR network, and analyzes/verifies their assumptions via various experiments. However, experimental validation of applications utilizing degradation-related representations is somewhat lacking. In particular, degradation-related representations should be combined to the recent SOTA  blind SR methods.

---

> ### Author Response · Authors · 2022-11-18
> **Response to Reviewer xxDc**
>
> We thank reviewer xxDc for his/her review time and comments. We discuss the reviewer's concerns one by one as follows:
>
> `Q1`: "Comparative analysis with existing blind SR techniques such as IKC is required, and experiments on other degradations such as compressing rather than noise and blur are also required."
>
> `A1`: Thanks for your suggestions. We add experiments to compare with IKC (CVPR19) and DAN (NIPS20). The results are as follows:
>
> | Method | Blur2 | Noise20 | Blur2+Noise20 |
> |---|---|---|---|
> | DASR (b+n) | 23.28 / 6.74 | 22.23 / 7.05 | 21.32 / 7.53 |
> | IKC (b) | 23.74 / 6.56 | 16.60 / 7.22 | 16.19 / 6.87 |
> | MANet (b+n) | 15.97 / 6.37 | 16.32 / 6.61 | 16.83 / 7.29 |
> | DAN (b+n) | 23.94 / 6.44 | 18.46 / 8.20 | 17.76 / 8.04 |
> | RRDB (clean) | 21.40 / 8.01 | 17 80 / 8.29 | 17.23 / 8.73 |
> | RRDB (b+n) | 23.79 / 6.36 | **22.54** / 6.66 | 21.36 / 7.36 |
> | RRDB-DDR (b+n) | **24.01** / **6.34** | 22.52 / **6.60** | **21.41** / **7.27** |
>
> `Q2`: "In the motivation section, experiments for observations are biased to image denoising (or restoration). Also, compared to the traditional method is BM3D which is a representative denoising method. When the reader first sees the paper, it seems strange. Figure 2 should be an experiment on SR."
>
> `A2`: Thanks for the comment. (1) CinCGAN is trained with degraded data in the motivation section. It performs SR with simultaneous restoration (denoise and deblur). If the input image's degradation is not included in the training data, CinCGAN will fail to transfer the degraded input to a clean one. More interestingly, instead of producing extra artifacts in the image, it seems that CinCGAN does not process the input image and retains all the original defects. In other words, the network seems to figure out the specific degradation types within its training data distribution, and distribution mismatch may make the network “turn off” its ability. For comparison, we adopt BM3D to process different types of noises. BM3D has an obvious and stable denoising performance for all different degradation types. The interesting phenomenon indicates that deep networks demonstrate the ability to distinguish among different degradation types. Hence, we study SR networks with degraded input images. That’s why the experiments are based on degraded images. (2) Based on the observation, in this paper, we disclose that a well-trained deep SR network is naturally a good descriptor of degradation information. Specially, we train an SR network with only clean LR images without any other degradations. Then, the deep representations of the trained network show discriminability to different degradations when testing. Note that the network does not perceive any degradation data during training.
>
> `Q3`: "Figure 2 does not show observations for various degradations. Not only noise, but also degradation such as compression, irregular holes, and color tones should be dealt with as well."
>
> `A3`: Thanks for the suggestion. In fact, the Hollywood dataset contains various degradations including non-Gaussian noise, compression, motion blur and other complex degradations. The DIV2K-mild dataset also includes diverse degradations like noise, blur, luminance jitter, etc. Further, in Section 5, we use DDR features to conduct image distortion identification. As described in Table 1, the LIVE dataset contains Gaussian Blur (GB), White Noise (WN), JPEG, JP2K, and Simulated Fast Fading Rayleigh Channel (FF) degradations. The results show that DDR features achieve excellent results in recognizing different distortion types. Therefore, we indeed verify our discovery on various degradations. The noise and blur degradations in Fig. 2 are adopted for simplicity and better illustration.
>
>
> `Q4`: "In addition, it is necessary to verify whether there is a different phenomenon that varies depending on the upsampling factor of the SR network."
>
> `A4`: In Section 4.5, we analyze the causes of DDR phenomenon. Concretely, the DDR phenomenon is mainly introduced by overfitting the degradation in the training data. Since the training data (DIV2K-clean) do not contain extra degradations, the trained SR network lacks the ability to deal with unseen degradations. Feeding images with degradations (e.g., noise and blur) will produce features with unprocessed noises or blurring. These patterned features naturally show a strong discriminability between different degradations. The global residual, adversarial training and longer training iteration all promote the DDR phenomenon. Therefore, the upscaling factor is not the essential determinant. It does not influence the main conclusion of this paper but is interesting to explore in the future.
>
>
> `Q5`: Minor points.
>
> `A5`: Thanks for pointing out these minor typos. We have revised our manuscript according to your comments.

---

> ### Author Response · Authors · 2022-11-22
> **Further discussion with Reviewer xxDC**
>
> Dear reviewer xxDC:
>
> We thank you for the precious review time and valuable comments. We have provided corresponding responses and results, which we believe have covered your concerns. We realized that we could no longer revise our manuscript. But we can continue the discussion to address your concerns.
>
> We hope to discuss further with you whether or not your concerns have been addressed. Please let us know if you still have any unclear parts of our work.
>
> Best,
>
> Paper 718 Authors.

---

### Official Review · Reviewer_EvNB · 2022-10-26

**Confidence:** 5
**Correctness:** 4
**Technical Novelty And Significance:** 4
**Empirical Novelty And Significance:** 4
**Recommendation:** 8

**Clarity, Quality, Novelty And Reproducibility:**

This work is well motivated and easy to follow. Extensive experiements are conducted to support their insightful analysis.

**Strength And Weaknesses:**

Pros:
Strength:
1 The paper is well-written and easy to read.
2 THis work is well motivated by the observation of applying pretrained SR models to images of different types of degradations. The idea about DDR is very interesting and seems novel to me.
3. Authors conduct extensive experiments to analyze the observation and give reasonable explanations.

Cons:
1 In this work CNN-based models such as SRResNet and SRGAN are adopted to illustrate the DDR generated by SR network. Currently Transformer-based SR network like SwinIR also achieves promising results. Does the observation about DDR also apply to the Transformer-based methods?
2. Do SR models with stronger SR performance perform better in indicating degradation? Can DDR also has the ability to indicate different levels of degradations?
3. In Fig. 3, why are training and test set for CinCGAN and SRGAN chosen to be different? It seems to me that the same setting is more meaningful.
4. In section5, injecting DDR embedding shows better performance, it is better to show some visual results for comparison to illusrate the advantage of DDR guidance.
5. DDR can indeed indicate the feature difference among the input with different degradations, however, is it reasonable to call DDR a kind of sematics?

**Summary Of The Paper:**

This paper proposes that SR network can learn the distinctive “semantics” named deep degradation representations (DDR). Considering through GAN-based models the SR network can generate more delicate SR textures, the authors believe that there are some kinds of semantic the SR network has learned. Through comprehensively analyzing the feature representations, the authors discover that the semantics in SR network is related to different degradations and two factors are adversarial learning and global residual. Experiments show that adversarial learning and global residual are important and DDR can be applied in many applications and achieves promising results.

**Summary Of The Review:**

Overall, I would like to have this work being accepted because of inspiring observation and insightful analysis.

---

> ### Author Response · Authors · 2022-11-18
> **Response to Reviewer EvNB (Part 2)**
>
> `Q3`: "In Fig. 3, why are training and test set for CinCGAN and SRGAN chosen to be different? It seems to me that the same setting is more meaningful."
>
> `A3`: Thanks for the question. We use CinCGAN and SRGAN as two representative methods for blind and classical SR, respectively. CinCGAN tackles unknown degradations and conducts experiments on DIV2K-mild. We illustrate that its deep representations have discriminability to different degradations with different datasets. Specifically, DIV2K-mild is the in-distribution training and testing dataset, while DIV2K-noise20 is synthesized with Gaussian noise level 20 and is perceptually similar to DIV2K-mild. These two datasets have the same image content but different degradations. Hollywood is a real-world old film dataset, and is used to demonstrate that the findings also hold true for real-world data. Fig. 3(a) clearly shows that the deep representations of CinCGAN have strong discriminability to these three degradations.
> SRGAN serves as an example of classical SR designed for fixed bicubic downsampling. Specifically, DIV2K-clean is in-distribution training and testing dataset, and we additionally consider blur and noise data as outside-distribution with the same image content as DIV2K-clean. The results are similar to CinCGAN that SRGAN can distinguish different degradations with the same image content.
>
>
> `Q4`: "In section5, injecting DDR embedding shows better performance, it is better to show some visual results for comparison to illustrate the advantage of DDR guidance. "
>
> `A4`: Thank you for your suggestions. Due to space limitations, we do not show visual examples. Following previous works, we present quantitative results in the paper, demonstrating the proposed method's effectiveness.
>
> `Q5`: "DDR can indeed indicate the feature difference among the input with different degradations, however, is it reasonable to call DDR a kind of semantics?"
>
> `A5`: We use the word ‘semantic’ to describe the ability of SR networks to distinguish between different degradations, which is in line with human perception of image distortions and could also serve as an analogy to high-level tasks like classification. In classification, the semantic refers to the human annotations on object classes, which are artificially defined for supervision. For SR, the observed semantics are distinct from those related to image content, but are closely related to image degradation types and can also be learned unsupervised without any human annotations. We hope this work could serve as a starting point for more research in network interpretability in low-level vision.

---

> ### Author Response · Authors · 2022-11-18
> **Response to Reviewer EvNB (Part 1)**
>
> Thank you so much for appreciating our work and the valuable comments. For your questions and concerns, we would like to respond as follows.
>
> `Q1`: "In this work CNN-based models such as SRResNet and SRGAN are adopted to illustrate the DDR generated by SR network. Currently Transformer-based SR network like SwinIR also achieves promising results. Does the observation about DDR also apply to the Transformer-based methods?"
>
> `A1`: In this paper, we use the most well-known SR networks, like SRResNet and SRGAN, to demonstrate our main observations and analysis. To justify the generality of DDR in terms of different building blocks, we also adopt the residual channel attention layer as the basic block, which is another popular choice for SR, as proposed in SENet and RCAN. Please kindly refer to Appendix A.9 for more details. Specifically, the corresponding observations are consistent with those of CNN-based methods. This suggests that the semantic representations of DDR do not stem from certain network structures, but from the SR task itself. As for Transformer-based methods for low-level vision, the transformer block is only adopted as the internal building block, just like the convolutional layer in SRResNet and the channel attention layer in RCAN. There are still convolutional layers in the front and back ends. Thus, its behaviors are predictable to be similar to CNN-based methods.
>
>
> `Q2-1`: "Do SR models with stronger SR performance perform better in indicating degradation?"
>
> `A2-1`: Thanks for the question. (1) We need to discuss this problem with two separate divisions of classical and blind SR:
>
> For classical SR with fixed bicubic downscaling, better SR performance may indicate a better indication of degradation if the networks are with the same structure. We have conducted experiments on SRCNN with different depths in Appendix A.6. Our experiments show that the model achieves better SR performance as well as better discriminability on degradation type with the increase of network depth. However, for networks with different structures, there is no clear relationship between SR performance and discriminability on degradation, as indicated by the results of SRCNN in Figure 13 of Appendix A.6 and SRResNet in Figure 6 of the main paper. Since different structures could differ a lot in terms of learned feature representations, the CHI metric is not bound to keep the same distribution.
>
> For blind SR tackling different degradation types, the answer is definitely NO. As discussed in Section 4.6 and Section 5 in the main paper, since SR networks tend to overfit degradation distributions, we need to diminish the feature discriminability to degradations in order to improve the model generalization for various degradations.
>
> `Q2-2`: "Can DDR also have the ability to indicate different levels of degradations?"
>
> `A2-2`: Thanks for the question. We have explored discriminability on degradation degrees in Appendix A.8, and the answer is YES. Specifically, we test SR networks on different noise and blur levels, and the results indicate that greater differences between degradation levels lead to stronger discriminability of feature representations.

---

> ### Author Response · Authors · 2022-11-22
> **Further discussion with Reviewer EvNB**
>
> Dear reviewer EvNB:
>
> We thank you for the precious review time and valuable comments. We have provided corresponding responses and results, which we believe have covered your concerns. We realized that we could no longer revise our manuscript. But we can continue the discussion to address your concerns.
>
> We hope to discuss further with you whether or not your concerns have been addressed. Please let us know if you still have any unclear parts of our work.
>
> Best,
>
> Paper 718 Authors.

---

### Official Review · Reviewer_CXK4 · 2022-10-26

**Confidence:** 5
**Correctness:** 2
**Technical Novelty And Significance:** 2
**Empirical Novelty And Significance:** 2
**Recommendation:** 3

**Clarity, Quality, Novelty And Reproducibility:**

The paper has an excellent writing and a clear idea. But I strongly doubt its significance to the related research.

**Strength And Weaknesses:**

[Strength]
1. The paper is well-written and easy to understand.
2. The paper provides comprehensive analyses, demonstrations, and discussions on DDR. The applications to blind SR, distortion identification and generalization evaluation are also interesting.

[Weakness]
1. The main concern is that the issue this paper explores would be trivial for further low-level research and realistic SR applications. There are three aspects that I doubt the significance of this work:
- Empirical settings: All the empirical demonstrations and the important observations and conclusions are mainly based on synthetic data, including artificial blur and noise. Particularly, it is strange that for noise, two version datasets are used for demonstrations, namely, DIV2K-noise20 and DIV2K-noise. For example, in Fig.3, subfigure (a)~(d) has different datasets.
- Although it claims Hollywood100 with real-world old film degradations, it has a stable intrinsic property, indicating film style, image degradation, etc. Even though it presents a different feature distribution from DIV2K-mild/-noise in Fig.3, it is not direct to draw the conclusion that this difference results from the image degradation. I do not think the analyses for DDR or "semantic" are convincing. Besides, why not use a more complex real-world SR dataset, e.g., RealSR, rather than Hollywood100?
- If ignoring the problems above, it assumes that the observed "semantics" hold. In this case, are those observations and conclusions important for the SR task, especially for real-world SR? The paper has few insights on how to indeed address SR problems. Although there are discussions on blind SR, the paper just conducts the evaluation on synthetic data and shows the learned features. What is the importance of the observation from Fig.11 (in Page 9, "Fig.11 reveals that DDR guidance can make the deep features become more homogenous")? Besides,  the demonstrations in Fig.11 are conducted on Urban100? it is also confusing. Urban100 is not mentioned in main paper.
2. On one hand, the demonstrations and analyses are based on very few SR methods with simple network architectures, for example, not including RCAN with attention mechanism or Transformer based networks. It would be misleading, confusing, and not convincing to understand the contributions of this paper, let alone do further SR research based on this paper. On the other hand, most of the experiments are conducted on synthetic data with artificial image degradation. It is ideal and hand-craft, showing few promising insights on further research.

**Summary Of The Paper:**

This paper aims to analyze the feature representations and explore the "semantics" in SR networks, namely, deep degradation representations. And authors reveal two factors, i.e., adversarial learning and global residual, which influence the extraction of "semantics".

**Summary Of The Review:**

Most of demonstrations are mainly based on synthetic data. It is not optimistic for its insights to the future work. Even it may bring the misleading to understand the SR research.

---

> ### Author Response · Authors · 2022-11-18
> **Response to Reviewer CXK4 (Part 2)**
>
> `Q3`: "Most of the experiments are conducted on synthetic data with artificial image degradation. It is ideal and hand-craft, showing few promising insights on further research."
>
> `A3`: Thanks for the question. However, we respectfully disagree with your comment. (1) The reason why we adopt the synthetic datasets is explained in `Q1`. The difficulty of the real-world datasets is that it is hard to keep the content the same but change the degradations. If we simply use two real-world datasets which contain different contents and different degradations, it is hard to say whether the feature discriminability is targeted at image content or at image degradation. Hence, synthetic data can at least control the variables.
>
> (2) However, following the reviewer's suggestions, we find a plausible real-world dataset Real-City100, which is proposed in the paper “Cameral SR”. The authors use iPhone X and NikonD5500 devices to capture controllable images. By adjusting the camera focal length, each camera captures paired images with the same content but different resolutions. The low-resolution images contain real-world degradations such as real noise and real blur. We test SRGAN on this dataset and obtain corresponding visualization results. The results are added in the revised version in Appendix A.3. It can be seen that the deep representations of SRGAN can still distinguish among different degradations across different devices.
>
> `Q4`: "Are those observations and conclusions important for the SR task, especially for real-world SR? ... What is the importance of the observation from Fig.11?"
>
> `A4`: Thanks for the question. We indeed draw important insights into the real-world and bind SR:
>
> (a) DDR can be used to measure the model generalization ability. In Section 4.6 and Appendix A.7, we illustrate that if the model is trained only on clean LR data, the deep representations show strong discriminability to clean and noisy data. In contrast, if the model sees noise data during training, such discriminability diminishes. The model will become more robust to more degradation types, as the distributions of the deep representations become unanimous. In summary, to improve the model generalization for various degradations, we need to diminish the feature discriminability to degradations. Thus, DDR can be used as an approximate evaluation metric for generalization ability. Specifically, we can obtain their DDR features given a trained model and several test datasets with different degradations. By evaluating the discriminability of the projection results (clustering effect), we can roughly measure the generalization performance over different degradation types. The worse the clustering effect, the better the generalizability. Fig .11 shows the DDR clustering of different models. In Fig. 11, RRDB (clean) is unable to deal with degraded data and obtains lower PSNR values on blur and noise inputs. Its CHI score is 322.16. By introducing degraded data into training, the model gains better generalization and the CHI score is 14.04. With DDR guidance, the generalization ability is further enhanced. The CHI score decreases to 4.95. The results are consistent with the results in the previous section. Interestingly, we do not need ground-truth images to evaluate the model generalization.
>
> (b) We propose to exploit DDR as guidance for the blind SR method. DDR guidance can help improve the model's performance. Fig. 11 reveals that DDR guidance can make the deep features become more homogeneous (CHI scores drop from 14.04 to 4.95). The above results both show the important value of DDR on generalization measurement and method design.
>
>
> `Q5`: "Besides, the demonstrations in Fig.11 are conducted on Urban100?"
>
> `A5`: Sorry for the missing explanation of Urban100. Urban100 is a widely-used dataset in the SR field. Almost all SR literature will evaluate the performance on the Urban100 dataset. It has been a common sense and default option in this field.
>
> `Q6`: "The demonstrations and analyses are based on very few SR methods with simple network architectures, for example, not including RCAN with attention mechanism or Transformer based networks."
>
> `A6`: We choose well-known structures to illustrate our findings. In addition, in Appendix A.9, we adopt the residual channel attention layer as the basic building block, which is inspired by SENet and RCAN. From the results, we can see that the observations are consistent with the findings in the main paper. It suggests that the semantic representations do not stem from network structures, but from the task itself. As for Transformer-based methods, in low-level vision, the transformer block is only adopted as the internal building block. There are still CNN layers in the front and back ends. Thus, its behaviors are predictable to be similar to CNN-based methods.

---

> ### Author Response · Authors · 2022-11-18
> **Response to Reviewer CXK4 (Part 1)**
>
> We thank reviewer CXK4 for his/her review time and comments. We discuss the reviewer's concerns one by one as follows:
>
> `Q1`: "All the empirical demonstrations and the important observations and conclusions are mainly based on synthetic data, including artificial blur and noise. Particularly, it is strange that for noise, two version datasets are used for demonstrations, namely, DIV2K-noise20 and DIV2K-noise. For example, in Fig.3, subfigure (a)~(d) has different datasets."
>
> `A1`: Thanks for the question. (1) We use synthetic data to control the content and degradation. To better show that DDR is more sensitive to image degradation rather than image content, we have to keep the content the same but change the degradation. The experimental results reveal that DDR is more degradation-related. It is hard to illustrate our findings fairly if we use real-world images with various degradations.
>
> (2) We use different datasets for different purposes. In Fig. 3 (a), we mainly explain that the deep representations of CinCGAN have discriminability to different degradations. Concretely, DIV2K-mild is the in-distribution training and testing dataset for CinCGAN. DIV2K-noise20 is synthesized with Gaussian noise level 20, which is perceptually similar to DIV2K-mild. They have the same image content but different degradations. Hollywood is a real-world old movie dataset used to demonstrate that findings about deep degradation of SR networks also apply to real-world data. Fig. 3(a) clearly shows that the deep representations of CinCGAN have strong discriminability to these three degradations. Fig. 3(b) is a comparison. We use a shallow 3-layer SRCNN trained on DIV2K-mild to conduct the same experiment. The results show that the deep features of SRCNN do not have a similar property to CinCGAN. We can conjecture that the degradation-related semantics only exist in deep models, not traditional methods or shallow networks. The above findings are the observations that drive us to explore the deep representations of SR networks. Fig. 3 (c) and (d) are comparison experiments between SRCNN and SRGAN. We additionally consider blur and clean data. The results are similar. Deeper SRGAN can distinguish different degradations with the same content, while shallower SRCNN is not able to discriminate degradations. The discovery is non-trivial.
>
> `Q2`: "Hollywood100 with real-world old film degradations has a stable intrinsic property, indicating film style, image degradation, etc. Even though it presents a different feature distribution from DIV2K-mild/-noise in Fig.3, it is not direct to draw the conclusion that this difference results from the image degradation. I do not think the analyses for DDR or "semantic" are convincing. Besides, why not use a more complex real-world SR dataset, e.g., RealSR, rather than Hollywood100?"
>
> `A2`: Thanks for the comments. (1) The reviewer's concern that real-world datasets have shortcomings is what confuses us, since the reviewer expressed concern in a previous question that real data was not used. We understand the reviewer's concern and we have considered this situation. As we explained before, real-world degraded images are hard to control the content. Thus, we first use synthetic data to control the content to be the same. The results show that even though the image content is the same, the deep representations still show strong discriminability to degradations. Hence, we conclude that the deep representations of SR networks are more sensitive to image degradation instead of image content. The conclusion is very clear and direct. Further, we also show the results on a real-world dataset. It is a reasonable choice. Both synthetic and real-world datasets are considered in our experiments. In addition, we conduct more experiments on the real-world dataset Real-City100. The details are illustrated in Q3. Please refer to it.
>
> (2) Hollywood has a stable intrinsic property, indicating film style, and image degradation. Isn’t it a good property? Our findings illustrate that DDR can distinguish different degradation types. Hollywood has a stable degradation type so we can more convincingly show the behaviors of DDR on this dataset. If one dataset includes various non-uniform degradations, it is hard to reveal the behavior of distinguishing degradation types. Thus, we choose a real-world dataset containing relatively consistent real-world degradations.
>
> (3) It is okay to adopt other real-world datasets. We choose the Hollywood dataset because this dataset is more representative of complex real-world old film degradations, which differs from simple image degradations like noise and blur. In terms of the complexity of degradation, Hollywood is more complex than RealSR. The experiments on Hollywood have already shown our findings.

---

> ### Author Response · Authors · 2022-11-22
> **Further discussion with Reviewer CXK4**
>
> Dear reviewer CXK4:
>
> We thank you for the precious review time and valuable comments. We have provided corresponding responses and results, which we believe have covered your concerns. We realized that we could no longer revise our manuscript. But we can continue the discussion to address your concerns.
>
> We hope to discuss further with you whether or not your concerns have been addressed. Please let us know if you still have any unclear parts of our work.
>
> Best,
>
> Paper 718 Authors.

---

### Official Review · Reviewer_DDMx · 2022-10-27

**Confidence:** 3
**Clarity, Quality, Novelty And Reproducibility:** The clarity, quality and novelty is a…
**Correctness:** 3
**Technical Novelty And Significance:** 3
**Empirical Novelty And Significance:** 3
**Recommendation:** 5

**Strength And Weaknesses:**

Strength:

1 Exploring the intrinsic representation of low-level vision models is of great importance to help the design of new methods. This paper makes a step forward towards this goal.
2 This paper is easy to read, some analysis are inspirable.

Weakness:

1 The key issue of this paper is how this empirical finding can facilitate the SR task. In common sense, the restoration of textures, especially those regular patterns with strong perceptual prior, relies on the semantic information, or at least the similar patterns that are included in the training set. However, according to the findings in this paper, the model mainly learns the degradation-related information. Is this good or bad for the performance of SR? or in other words if the authors recommend the following researches to keep the global residual and the adversarial loss in the designing of SR network?
2 There lacks sufficient experiments to verify the performance of blind SR with the proposed DDR guidance. Detailed comparison against the state-of-the-art methods should be conducted.


**Summary Of The Paper:**

This paper discovers an interesting phenomenon that the SR network is powerful in discriminating the image degradations instead of image contents, especially for well-trained deep networks with global residual and generative adversarial training. To validate the claim, the authors give some empirical evidence by visualizing the distribution of features via t-SNE. In addition, some inspirable analysis and applications are provided and reported.

**Summary Of The Review:**

This paper reports an interesting phenomenon that the SR network shows more discriminative capacity in degradation types rather than semantics. While some analysis are inspirable, the authors fails to validate its effectiveness in facilitating the performance of the SR task.

---

> ### Author Response · Authors · 2022-11-17
> **Response to Reviewer DDMx**
>
> We thank reviewer DDMx for his/her review time and comments. We discuss the reviewer's concerns one by one as follows:
>
> `Q1`. "The key issue of this paper is how this empirical finding can facilitate the SR task. According to the findings in this paper, the model mainly learns the degradation-related information. Is this good or bad for the performance of SR? or in other words if the authors recommend the following researches to keep the global residual and the adversarial loss in the designing of SR network?"
>
> `A1`: Thanks for the question. (1) In this paper, the main contribution is to explore and discover the semantics of deep degradation representations (DDR) of SR networks, i.e., a well-trained SR network is naturally a good descriptor of image degradation. The purpose of this paper is not restricted to facilitating SR task performance. The discovery is essential for understanding the network mechanisms. We do apply our findings to several fundamental tasks and achieve promising results, but the main contribution of this paper does not lie in the application section. As an analytical study, understanding deep SR networks is as important as proposing a specific algorithm. (2) The findings are discovered by classical SR networks, which means the training data do not contain extra blur or noise degradations. Thus, our findings do not influence the performance of classical SR, because classical SR assumes the downsampling kernel is bicubic without any further degradation. (3) For blind SR (input images contain additionally unknown degradations), our findings can facilitate the method design. Specifically, we show the potential of well-trained SR networks as degradation descriptors that can separate different degradation types. Hence, we can exploit a trained SR network to extract degradation information, and use such information as additional prior to support a blind SR network. We conduct this experiment In Section 5. The results show DDR can improve performance, validating the significance of our discovery. Besides, we have explored more applications like distortion identification. The results also show a positive signal of exploiting the potential of DDR.
>
> `Q2`: "There lacks sufficient experiments to verify the performance of blind SR with the proposed DDR guidance. Detailed comparison against the state-of-the-art methods should be conducted."
>
> `A2`: Thanks for your comment. We exploit DDR as prior guidance to facilitate the SR networks. It is a plug-and-play design that can be flexibly incorporated into various backbones. In the experiment, we adopt RRDBNet as the SR backbone, since RRDBNet is widely adopted as the backbone, including BSRGAN, Real-ESRGAN, etc. The results show that DDR guidance can help improve the model generalization. We compare with DASR (CVPR21) in Tab. 2. Following your suggestion, we also compare the performance of MANet (ICCV21). We use their released codes and pre-trained models to test.
>
> | Method         | Blur2                | Noise20          | Blur2+Noise20        |
> |----------------|----------------------|------------------|----------------------|
> | DASR (b+n)     | 23.28 / 6.74         | 22.23 / 7.05     | 21.32 / 7.53         |
> | IKC (b)        | 23.74 / 6.56         | 16.60 / 7.22     | 16.19 / 6.87         |
> | MANet (b+n)    | 15.97 / 6.37         | 16.32 / 6.61     | 16.83 / 7.29         |
> | DAN (b+n)      | 23.94 / 6.44         | 18.46 / 8.20     | 17.76 / 8.04         |
> | RRDB (clean)   | 21.40 / 8.01         | 17 80 / 8.29     | 17.23 / 8.73         |
> | RRDB (b+n)     | 23.79 / 6.36         | **22.54** / 6.66 | 21.36 / 7.36         |
> | RRDB-DDR (b+n) | **24.01** / **6.34** | 22.52 / **6.60** | **21.41** / **7.27** |
>
> For MANet, we find that it tends to produce results with pixel offsets (misalignment with GT), leading to low PSNR values. However, its NIQE values are good. The proposed DDR guidance can be further plugged into MANet to achieve better performance.
>
> `Additional Comment`:
>
> We notice that the reviewer thinks "Some of the paper’s claims have minor issues". We would like to ask if there are any issues other than those discussed above. If the reviewers can point it out, we can explain and discuss it further in the following discussion. We hope to address any concerns of the reviewer.

---

> > ### Comment · Reviewer_DDMx · 2022-12-09
> > **Thanks for your detailed responses.**
> >
> > Thanks for your detailed responses, and I have read the comments from other reviewers as well as your respective responses. Exploring the network mechanisms is indeed a good contribution of this work. However, I still think the main findings about the degradation-related learning of an image restoration network need further validation via extensive experimental results. The experiments in A2 show some insights, yet are restricted to very specific degradations and datasets, as well as metrics that can hardly reflect perceptual quality well. In general, the restoration network may fit some degradation spaces in a specific senerio, yet the texture-related or content-related information should also be used to restore image patterns. Given the above reasons, I maintain my original recommendation.

---

> ### Author Response · Authors · 2022-11-22
> **Further discussion with Reviewer DDMx**
>
> Dear reviewer DDMx:
>
> We thank you for the precious review time and valuable comments. We have provided corresponding responses and results, which we believe have covered your concerns. We realized that we could no longer revise our manuscript. But we can continue the discussion to address your concerns.
>
> We hope to discuss further with you whether or not your concerns have been addressed. Please let us know if you still have any unclear parts of our work.
>
> Best,
>
> Paper 718 Authors.

---

### Decision · Program_Chairs · 2023-01-20

**Decision:**

Reject

**Justification For Why Not Higher Score:**

The paper studies only limited forms of degradation. I believe a more comprehensive study of architectures and degradation, coupled with real data would make the paper stronger. The key observation in this paper about DDR is interesting and warrants further study.

**Justification For Why Not Lower Score:**

N/A

**Metareview: Summary, Strengths And Weaknesses:**

*Summary*: The paper presents an understanding of Super Resolution networks and their learned representations.  Their main finding is that the super resolution networks learn deep degradation representations (DDR).

*Strengths*: (1) This paper has an interesting finding about SR networks, namely the DDR. The DDR can be applied to many downstream tasks such as blind SR, distortion identification etc. (2) The connection of the feature quality to adversarial learning and global residual is interesting. (3) Good experimental evaluation. The authors also responded to the reviewers and included comparisons to other recent methods like IKC.

*Weaknesses*: (1) The main types of degradation studied in this paper are synthetic and limited to noise based perturbations. (2) The study is limited to only certain architecture types. As reviewers EvNB & CXK4 point out, recent Transformer based architectures are not studied in this work.